# A beta-glucosidase of an insect herbivore determines both toxicity and deterrence of a dandelion defense metabolite

Meret Huber[1,2]*, Thomas Roder[3†], Sandra Irmisch[2‡], Alexander Riedel[2§], Saskia Gablenz[2], Julia Fricke[3], Peter Rahfeld[4#], Michael Reichelt[2], Christian Paetz[5], Nicole Liechti[3¶], Lingfei Hu[3**], Zoe Bont[3], Ye Meng[3††], Wei Huang[3‡‡], Christelle AM Robert[3], Jonathan Gershenzon[2], Matthias Erb[3]*

[1]Institute of Plant Biology and Biotechnology, University of Muenster, Muenster, Germany; [2]Department of Biochemistry, Max-Planck Institute for Chemical Ecology, Jena, Germany; [3]Institute of Plant Sciences, University of Bern, Bern, Switzerland; [4]Department of Bioorganic Chemistry, Max-Planck Institute for Chemical Ecology, Jena, Germany; [5]Research group Biosynthesis/NMR, Max-Planck Institute for Chemical Ecology, Jena, Germany

*For correspondence: huberm@uni-muenster.de (MH); matthias.erb@ips.unibe.ch (ME)

Present address: †Interfaculty Bioinformatics Unit, University of Bern, Bern, Switzerland; ‡Michael Smith Laboratories, University of British Columbia, Vancouver, Canada; §Ernst Abbe University of Applied Sciences, Jena, Germany; #Department of Chemistry, University of British Colombia, Vancouver, Canada; ¶Institute for Infectious Diseases, University of Bern, Bern, Switzerland; **Institute of Soil and Water Resources and Environmental Sciences, Zhejiang University, Hangzhou, China; ††Tea Research Institute, Chinese Academy of Agriculture Sciences, Hangzhou, China; ‡‡CAS Key Laboratory of Aquatic Botany and Watershed Ecology, Wuhan Botanical Garden, Chinese Academy of Sciences, Wuhan, China

Competing interest: The authors declare that no competing interests exist.

**ABSTRACT** Gut enzymes can metabolize plant defense compounds and thereby affect the growth and fitness of insect herbivores. Whether these enzymes also influence feeding preference is largely unknown. We studied the metabolization of taraxinic acid β-D-glucopyranosyl ester (TA-G), a sesquiterpene lactone of the common dandelion (*Taraxacum officinale*) that deters its major root herbivore, the common cockchafer larva (*Melolontha melolontha*). We have demonstrated that TA-G is rapidly deglucosylated and conjugated to glutathione in the insect gut. A broad-spectrum *M. melolontha* β-glucosidase, Mm_bGlc17, is sufficient and necessary for TA-G deglucosylation. Using cross-species RNA interference, we have shown that Mm_bGlc17 reduces TA-G toxicity. Furthermore, Mm_bGlc17 is required for the preference of *M. melolontha* larvae for TA-G-deficient plants. Thus, herbivore metabolism modulates both the toxicity and deterrence of a plant defense compound. Our work illustrates the multifaceted roles of insect digestive enzymes as mediators of plant-herbivore interactions.

## Introduction

Plants produce an arsenal of toxic secondary metabolites, many of which protect them against phytophagous insects by acting as toxins, digestibility reducers, repellents, and deterrents (*Mithöfer and Boland, 2012*). Insect herbivores commonly metabolize defense metabolites, with important consequences for the toxicity of the compounds (*Heckel, 2014*; *Pentzold et al., 2014*). Recent studies identified a series of enzymes that metabolize plant defense metabolites and thereby benefit herbivore growth and fitness (*Sun et al., 2019*; *Sun et al., 2020*; *Poreddy et al., 2015*). However, to date, the behavioral consequences of insect metabolism of plant defense metabolites are little understood, despite the importance of behavioral effects of plant defenses for plant fitness and evolution in nature (*Mithöfer and Boland, 2012*; *Huber et al., 2016a*; *Huber et al., 2016b*; *War et al., 2012*).

Insect enzymes that were identified to metabolize plant defense compounds belong mainly to a few large enzyme classes including the cytochrome P450 monooxygenases, UDP-glycosyltransferases, and glutathione S-transferases (*Mao et al., 2007*; *Heidel-Fischer and Vogel, 2015*; *Bass et al., 2013*; *Maag et al., 2014*; *Wouters et al., 2014*). However, members of other enzyme groups can participate in detoxification, some of which are also involved in primary digestive processes for the breakdown

**eLife digest** Plants produce certain substances to fend off attackers like plant-feeding insects. To stop these compounds from damaging their own cells, plants often attach sugar molecules to them. When an insect tries to eat the plant, the plant removes the stabilizing sugar, 'activating' the compounds and making them toxic or foul-tasting. Curiously, some insects remove the sugar themselves, but it is unclear what consequences this has, especially for insect behavior.

Dandelions, *Taraxacum officinale*, make high concentrations of a sugar-containing defense compound in their roots called taraxinic acid β-D-glucopyranosyl ester, or TA-G for short. TA-G deters the larvae of the Maybug – a pest also known as the common cockchafer or the doodlebug – from eating dandelion roots. When Maybug larvae do eat TA-G, it is found in their systems without its sugar. However, it is unclear whether it is the plant or the larva that removes the sugar. A second open question is how the sugar removal process affects the behavior of the Maybug larvae.

Using chemical analysis and genetic manipulation, Huber et al. investigated what happens when Maybug larvae eat TA-G. This revealed that the acidity levels in the larvae's digestive system deactivate the proteins from the dandelion that would normally remove the sugar from TA-G. However, rather than leaving the compound intact, larvae remove the sugar from TA-G themselves. They do this using a digestive enzyme, known as a beta-glucosidase, that cuts through sugar. Removing the sugar from TA-G made the compound less toxic, allowing the larvae to grow bigger, but it also increased TA-G's deterrent effects, making the larvae less likely to eat the roots.

Any organism that eats plants, including humans, must deal with chemicals like TA-G in their food. Once inside the body, enzymes can change these chemicals, altering their effects. This happens with many medicines, too. In the future, it might be possible to design compounds that activate only in certain species, or under certain conditions. Further studies in different systems may aid the development of new methods of pest control, or new drug treatments.

of carbohydrates (β-glucosidases), proteins (proteases), and lipids (lipases). For instance, a *Manduca sexta* β-glucosidase deglycosylates the *Nicotiana attenuata* diterpene glycoside lyciumoside IV, thus alleviating its toxicity (***Poreddy et al., 2015***). Similarly, the Mexican bean weevil (*Zabrotes subfasciatus*) expresses a protease that degrades α-amylase inhibitors from its host, the common bean (*Phaseolus vulgaris*) (***Ishimoto and Chrispeels, 1996***). Finally, several insects degrade antinutritional plant protease inhibitors through intestinal proteases (***Giri et al., 1998***; ***Zhu-Salzman and Zeng, 2015***). Together, these studies suggest that families of typical digestive enzymes should be examined more carefully for possible roles in the detoxification of plant chemicals.

Enzymes involved in carbohydrate digestion may play a particular role in processing plant defense glycosides. Such compounds are typically considered protoxins, non-toxic, glycosylated precursors that are brought into contact with compartmentalized plant glycosidases upon tissue damage to yield toxic aglycones (***Wittstock and Gershenzon, 2002***). Both plant and insect glycosidases may activate plant defense glycosides (***Pentzold et al., 2014***). The alkaloid glucoside vicine in fava beans, for instance, is hydrolyzed to the toxic aglycone divicine in the gut of bruchid beetles (***Desroches et al., 1997***). Similarly, phenolic glycoside toxins are hydrolyzed rapidly by *Papilio glaucus*, the eastern tiger swallowtail. *P. glaucus* subspecies adapted to phenolic glycoside-containing poplars and willow show significantly lower hydrolysis of these metabolites (***Lindroth, 1988***). Finally, iridoid glycosides from *Plantago* species are hydrolyzed and thereby activated by herbivore-derived β-glucosidases, and β-glucosidase activity is negatively correlated with host plant adaptation both within and between species (***Pankoke et al., 2012***; ***Pankoke and Dobler, 2015***). These studies show that herbivore-derived enzymes may cleave plant protoxins and so may be a target of host plant adaptation. However, the genetic basis of protoxin activation by herbivores and the biological consequences of this phenomenon for insect feeding preference and performance are poorly understood.

Although the deglycosylation of plant defense metabolites is commonly assumed to be disadvantageous for the herbivore, a recent study in *M. sexta* showed that deglycosylation of a plant glycoside may decrease rather than increase toxicity (***Poreddy et al., 2015***). Silencing *M. sexta* β-glucosidase one resulted in developmental defects in larvae feeding on *N. attenuata* plants producing the diterpene glycoside lyciumoside IV (Lyc4), but not in larvae feeding on Lyc4-deficient plants, suggesting

that deglycosylation detoxifies rather than activates Lyc4. Although Lyc4 is an atypical defensive glycoside that carries several different sugar moieties and is only partially deglycosylated by *M. sexta*, these results bring up the possibility that defensive activation by glycoside hydrolysis does not necessarily increase the toxicity of these compounds, but may be a detoxification strategy. Clearly, more research on how glycoside hydrolysis by digestive enzymes impacts herbivores is needed to understand the role of this process in plant-herbivore interactions (*Pentzold et al., 2014*; *Poreddy et al., 2015*; *Marti et al., 2013*).

The herbivore toxins derived from glycoside protoxins have often been investigated for their defensive roles in connection with herbivore growth and development (*Poreddy et al., 2015*; *Desroches et al., 1997*; *Lindroth, 1988*; *Pankoke et al., 2012*; *Pankoke and Dobler, 2015*) rather than feeding deterrence, despite the fact that the latter is a well-established mechanism for plant protection in this context (*Pollard, 1992*). For example, the maize benzoxazinoid glucoside HDMBOA-Glc reduces food intake by *Spodoptera* caterpillars as soon as the glucoside moiety is cleaved off by plant β-glucosidases (*Glauser et al., 2011*). Similarly, the deterrent effect of cyanogenic glucosides in *Sorghum* toward *Spodoptera frugiperda* is directly dependent on a functional plant β-glucosidase that releases cyanide upon tissue disruption (*Krothapalli et al., 2013*). Furthermore, different glucosinolate breakdown products have been shown to affect oviposition and feeding choices by *Pieris rapae* and *Trichoplusia ni* (*de Vos et al., 2008*; *Zhang et al., 2006*; *Mumm et al., 2008*). However, whether protoxin activation by herbivore-derived enzymes influences herbivore host plant choice remains unknown.

All protoxin-activating enzymes that have been characterized so far in insect herbivores are β-glucosidases, which cleave β-D-glucosides and release free glucose (*Pentzold et al., 2014*). The primary role of β-glucosidases in insect digestion is to function in the last steps of cellulose and hemicellulose breakdown by converting cellobiose to glucose (*Zhang et al., 2012*). Most insect β-glucosidases, however, also accept other substrates, including various di- and oligosaccharides, glycoproteins, and glycolipids, which may help herbivores to obtain glucose from various sources and enable the further breakdown of glycosylated proteins and lipids (*Marana et al., 2000*; *Azevedo et al., 2003*; *Ferreira et al., 2003*; *Ferreira et al., 2001*). However, the broad substrate specificity of insect β-glucosidases for plant glucosides with an aryl or alkyl moiety may also result in the activation of defense metabolites, as discussed above (*Terra and Ferreira, 1994*). Thus, investigating the substrate specificity and the biochemical function of insect β-glucosidases is important to understand the ecology and evolution of insect-mediated protoxin activation.

Known plant protoxins include glucosinolates, salicinoids, and cyanogenic, iridoid, and benzoxazinoid glycosides. Plants produce many other types of glycosides that may also be protoxins, but most of these have not yet been carefully investigated for their toxicity or metabolic stability in herbivores. Among these potential protoxins are the bitter-tasting sesquiterpene lactone glycosides. Sesquiterpene lactones form a large group of over 2000 plant defense compounds found principally in the Asteraceae family, with glycosides especially common in the latex-producing tribe Cichorieae, which enters the human diet through lettuce, endive, and chicory (*Chadwick et al., 2013*). These substances have a long appreciated role in defense against insect herbivores (*Picman, 1986*), but it is not clear if glycosylated sesquiterpene lactones should be considered as protoxins that are activated by plant damage.

Here, we studied the metabolism of a sesquiterpene lactone glucoside during the interaction between the common dandelion *Taraxacum officinale* aggregate (Asteraceae, Chicorieae) and the larvae of the common cockchafer, *Melolontha melolontha* (Coleoptera, Scarabaeidae) (*Keller et al., 1986*; *Hasler, 1986*). *M. melolontha* larvae feed on roots of different plant species including members of Poaceae, Brassicaceae, Salicaceae, and Asteraceae families, which can contain glycosylated defense compounds such as benzoxazinoids, glucosinolates, and salicinoids, as well as sesquiterpene lactone glycosides (*Kondor et al., 2007*; *Hauss and Schütte, 1976*; *Hauss, 1975*; *Sukovata et al., 2015*). The alkaline gut pH of *M. melolontha* (pH = 8.0–8.5) possibly facilitates its polyphagous feeding habit by inhibiting the often acidic activating glucosidases of plant protoxins (*Pentzold et al., 2014*; *Egert et al., 2005*). In the third and final instars, *M. melolontha* prefers to feed on *T. officinale*, which produces large quantities of latex in its roots (*Hauss and Schütte, 1976*; *Huber et al., 2015*). The most abundant latex compound, the sesquiterpene lactone glucoside taraxinic acid (TA) β-D-glucopyranosyl ester (TA-G), deters *M. melolontha* feeding and thereby benefits plant fitness (*Huber et al., 2016a*; *Huber et al., 2016b*; *Huber et al., 2015*).

To understand the interaction between TA-G and *M. melolontha*, we first investigated whether TA-G is deglucosylated during insect feeding and whether plant or insect enzymes are involved. We then identified *M. melolontha* β-glucosidases that might hydrolyze TA-G through a comparative transcriptomic approach and narrowed down the list of candidate genes through in vitro characterization of heterologously expressed proteins. Finally, we silenced TA-G-hydrolyzing β-glucosidases in *M. melolontha* through RNA interference (RNAi) and determined the effect of these enzymes on TA-G hydrolysis, toxicity, and deterrence in vivo. Taken together, our results reveal that β-glucosidases modify the effects of plant defense metabolites on both herbivore performance and host plant choice, with potentially important consequences for the ecology and evolution of plant-herbivore interactions.

## Results

### TA-G is deglucosylated and conjugated to GSH during *M. melolontha* feeding

To test if TA-G is hydrolyzed during *M. melolontha* feeding, we analyzed larvae that had ingested defined amounts of TA-G-containing *T. officinale* latex. The aglycone TA was not detected in the latex itself but was present in substantial amounts in the regurgitant and gut of latex-fed larvae. TA-G on the other hand disappeared as soon as the latex was ingested by the larvae (*Figure 1A*). TA-glutathione (TA-GSH) and TA-cysteine (TA-Cys) were also identified in latex-fed larvae based on mass spectral and nuclear magnetic resonance (NMR) data, with the Cys sulfhydryl moiety being conjugated to TA at the exocyclic methylene group of the α-methylene-γ-lactone moiety (*Figure 1B–C*, *Figure 1—figure supplements 1–6*). Lower amounts of TA-Cys-Glu and TA-Cys-Gly were also present (*Figure 1—figure supplement 1*). No TA-G-GSH or TA-G-Cys conjugates were detected in this experiment. Based on current knowledge of the GSH pathway in insects (*Schramm et al., 2012*), it is likely that TA is first conjugated to GSH and then cleaved sequentially to form the other metabolites, although some conjugation to GSH prior to deglucosylation may also occur (*Figure 1C*). Quantitative measurements showed that approximately 25 % of the ingested TA-G was converted to GSH conjugates and derivatives (*Figure 1D*), with TA-Cys accounting for 95 % of all identified compounds (*Figure 1E*). TA-Cys mainly accumulated in the anterior midgut (*Figure 1D*), and this pattern was stable over prolonged exposure of *M. melolontha* to TA-G (*Figure 1—figure supplement 7*). In contrast to the different body parts, the frass only contained a small fraction of TA conjugates and was dominated by trace quantities of intact TA-G (*Figure 1D–E*). Thus, the deglucosylation and GSH conjugation of TA is a major route for metabolism of this sesquiterpene lactone in *M. melolontha*.

### Insect rather than plant enzymes catalyze TA-G deglucosylation in *M. melolontha*

TA-G deglucosylation may be mediated by plant or insect enzymes or a combination of both. TA-G in *T. officinale* latex incubated at different pH levels at room temperature was readily enzymatically deglucosylated to TA at a pH of 4.6 and 5.4, but not at lower or higher pH values (*Figure 2A*). As the midgut pH of *M. melolontha* is above 8 (*Figure 2B*; *Egert et al., 2005*), the deglucosylation of TA-G by plant-derived enzymes is likely inhibited. To test whether TA-G is hydrolyzed by *M. melolontha* enzymes, various *M. melolontha* gut sections were dissected and extracted. Strong deglucosylation activity was detected in the proximal parts of the gut, especially in the anterior midgut (*Figure 2B*). TA-G hydrolysis also occurred when larvae were fed with a diet containing heat-deactivated latex, which no longer hydrolyzes TA-G itself (*Figure 2A and C*), and the presence of TA-G-hydrolyzing latex proteins in TA-G-containing diets did not result in higher amounts of TA or TA conjugates inside *M. melolontha* compared to diets with heat-deactivated latex proteins (*Figure 2C*). Therefore, insect-derived enzymes are sufficient for TA-G deglucosylation in *M. melolontha*.

### TA-G hydrolysis is catalyzed by *M. melolontha* b-glucosidases

As the glucose moiety of TA-G is attached through an ester rather than a glycoside linkage, carboxylesterases or glucosidases may deglucosylate TA-G. TA-G deglucosylation by *M. melolontha* midgut protein extracts was inhibited by the addition of the α- and β-glucosidase inhibitor castanospermine in a dose-dependent manner, but not by the α-glucosidase inhibitor acarbose or the carboxylesterase

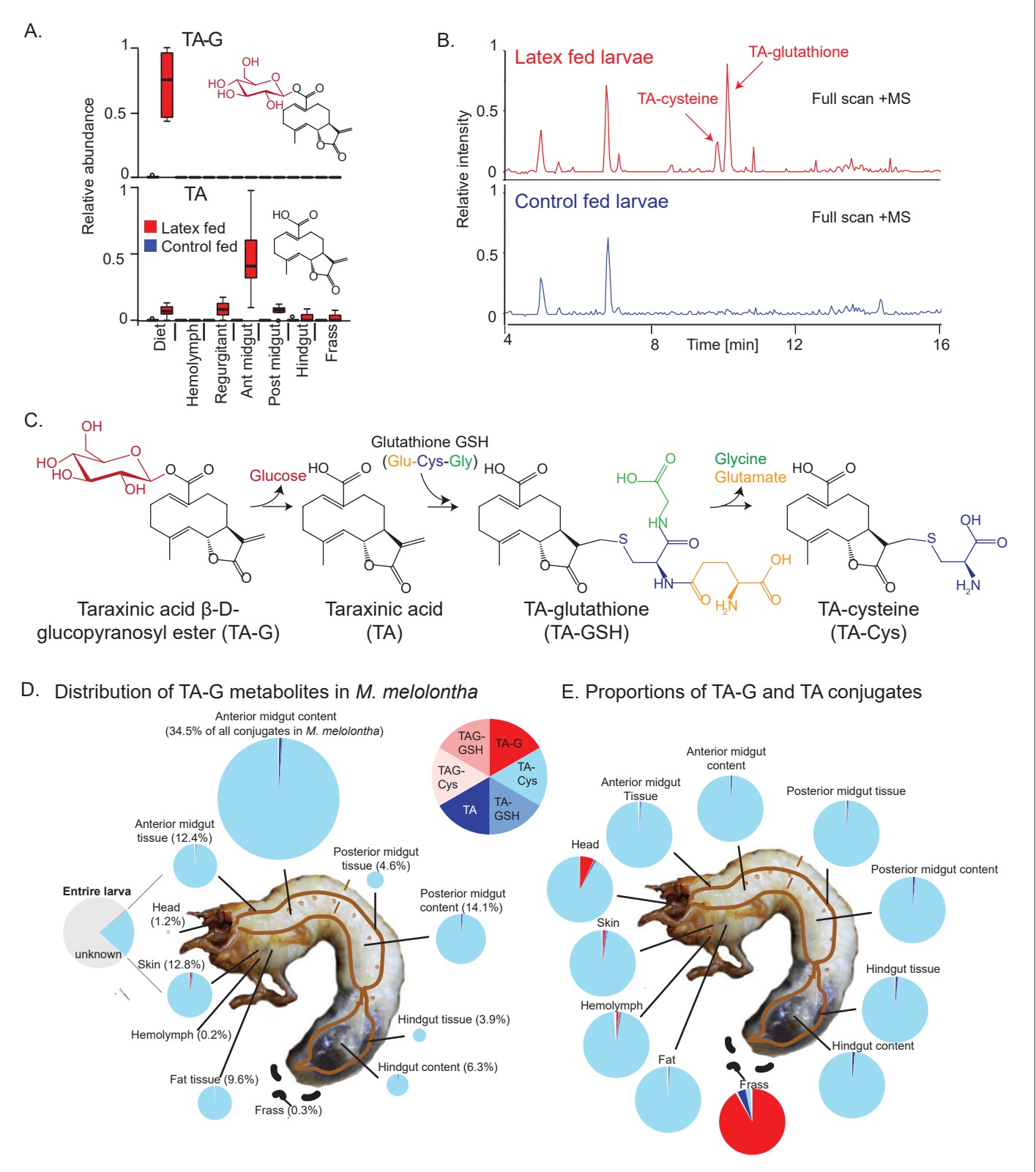

**Figure 1.** Taraxinic acid $\beta$-D-glucopyranosyl ester is rapidly deglucosylated and conjugated to glutathione upon ingestion by *Melolontha melolontha*. (**A**) Relative abundance of taraxinic acid $\beta$-D-glucopyranosyl ester (TA-G) and its aglycone taraxinic acid (TA) in diets enriched with *Taraxacum officinale* latex and in *Melolontha melolontha* larval gut, hemolymph, and frass after feeding on latex-containing and control diets. Ant = anterior; post = posterior. N = 5. For relative quantification of TA-glutathione (TA-GSH) conjugates in *M. melolontha*, refer to **Figure 1—figure supplement 1**. (**B**) High-

*Figure 1 continued on next page*

*Figure 1 continued*

pressure liquid chromatography-mass spectrometry (HPLC-MS) full scan (positive mode) of the anterior midgut of *M. melolontha* larvae fed with latex-containing and control diets. (**C**) Schematic illustration of proposed TA-G metabolism in *M. melolontha*. For nuclear magnetic resonance (NMR) analysis of TA-GSH conjugates, refer to *Figure 1—figure supplements 2–6*. Distribution of the total deglucosylated and conjugated metabolites of TA-G in *M. melolontha* larvae that consumed 100 µg TA-G within 24 hr. The size of the circles is relative to the total amount of conjugates. Values denote the percentage of metabolites found in each body part and are the mean of eight replicates. For long-term distribution of TA-Cys in *M. melolontha*, refer to *Figure 1—figure supplement 7*. (**E**) Relative proportions of TA-G metabolites in quantities from panel (**D**). Values denote the mean of eight replicates. Raw data are available in *Figure 1—source data 1*.

The online version of this article includes the following figure supplement(s) for figure 1:

**Source data 1.** Source data of main and supplementary figures of *Figure 1*.

**Figure supplement 1.** Relative quantification of TA-glutathione conjugates in *Melolontha melolontha* larvae feeding on diets with and without *Taraxacum officinale* latex.

**Figure supplement 2.** 500 MHz $^1$H-$^{13}$C HSQC NMR spectrum of the partially purified *Melolontha melolontha* midgut extract.

**Figure supplement 3.** Structure of synthesized TA-G-GSH with chemical shifts (500 MHz, in MeOD-$d_4$).

**Figure supplement 4.** Structure of synthesized TA-G-Cys with chemical shifts (500 MHz, in MeOD-$d_4$).

**Figure supplement 5.** Structure of synthesized TA-GSH with chemical shifts (500 MHz, in MeOD-$d_4$).

**Figure supplement 6.** Structure of synthesized TA-Cys with chemical shifts (500 MHz, in MeOD-$d_4$).

**Figure supplement 7.** Accumulation of TA-Cys in different body parts of *Melolontha melolontha* upon feeding for 1 month on *Taraxacum officinale* plants.

inhibitor bis(p-nitrophenyl)phosphate (*Figure 2—figure supplements 1–2*). This suggests that β-glucosidases rather than carboxylesterases catalyze TA-G deglucosylation in *M. melolontha*.

## Identification of gut-expressed *M. melolontha* b-glucosidases

In order to identify TA-G-hydrolyzing β-glucosidases, we separately sequenced 18 mRNA samples isolated from anterior and posterior midguts of larvae that had been feeding on diets coated with crude latex, TA-G-enriched extracts, or water. Putative *M. melolontha* β-glucosidases were identified based on amino acid similarity to known β-glucosidases from *Tenebrio molitor* and *Chrysomela populi*. 19 sequences similar to β-glucosidases had an expression profile matching the observed pattern of high TA-G deglucosylation activity in the anterior midgut. Partial sequences were extended using rapid-amplification of complementary DNA (cDNA) ends polymerase chain reaction (RACE PCR), resulting in 12 full-length β-glucosidases sharing between 55 and 79% amino acid similarity (*Figure 3A*, *Figure 3—figure supplement 1*, *Supplementary file 1*). The remaining seven transcripts could not be amplified or turned out to be fragments of the other candidate genes. All amplified sequences contained an N-terminal excretion signal and possessed the ITENG and NEP motifs characteristic of glucosidases (*Figure 3—figure supplement 1*; *Sanz-Aparicio et al., 1998*; *Davies and Henrissat, 1995*; *Barrett et al., 1995*). Expression levels of the candidate genes were 37- to 308-fold higher in the anterior than posterior midgut samples ($p_{adj}$ <10$^{-5}$, exact tests, n = 3), thus matching the differences in TA-G deglucosylation rate between these gut compartments (*Figure 3B*). Average expression of the transcripts did not differ among *M. melolontha* larvae fed water, TA-G, or latex (*Figure 3B*; $p_{adj}$ >0.50, exact tests, n = 3).

## Five *M. melolontha* b-glucosidases exhibit TA-G-hydrolyzing activity

The amplified *M. melolontha* β-glucosidases were heterologously expressed in an insect cell line and assayed with a variety of plant glycosides, including TA-G, benzoxazinoids, a salicinoid, and a glucosinolate as well as the disaccharide cellobiose. 9 of the 12 β-glucosidases were active with the standard fluorogenic substrate, 4-methylumbelliferyl-β-D-glucopyranoside, and hydrolyzed at least one of the plant metabolites (*Figure 3C*, *Figure 3—figure supplement 2*). For the three remaining enzymes, we did not observe hydrolysis of any substrate. Absence of any enzymatic activity could either be the result of a lack of catalytic activity toward the tested substrates or of low transgene expression and protein secretion by the cell line. All tested substrates were deglucosylated by at least one *M. melolontha* glucosidase (*Figure 3C*) in agreement with the hydrolysis activity of crude midgut extracts (*Figure 3—figure supplement 3*). Five heterologously expressed proteins deglucosylated TA-G (*Figure 3C*), with the highest TA aglycone formation found for Mm_bGlc17 (*Figure 3—figure*

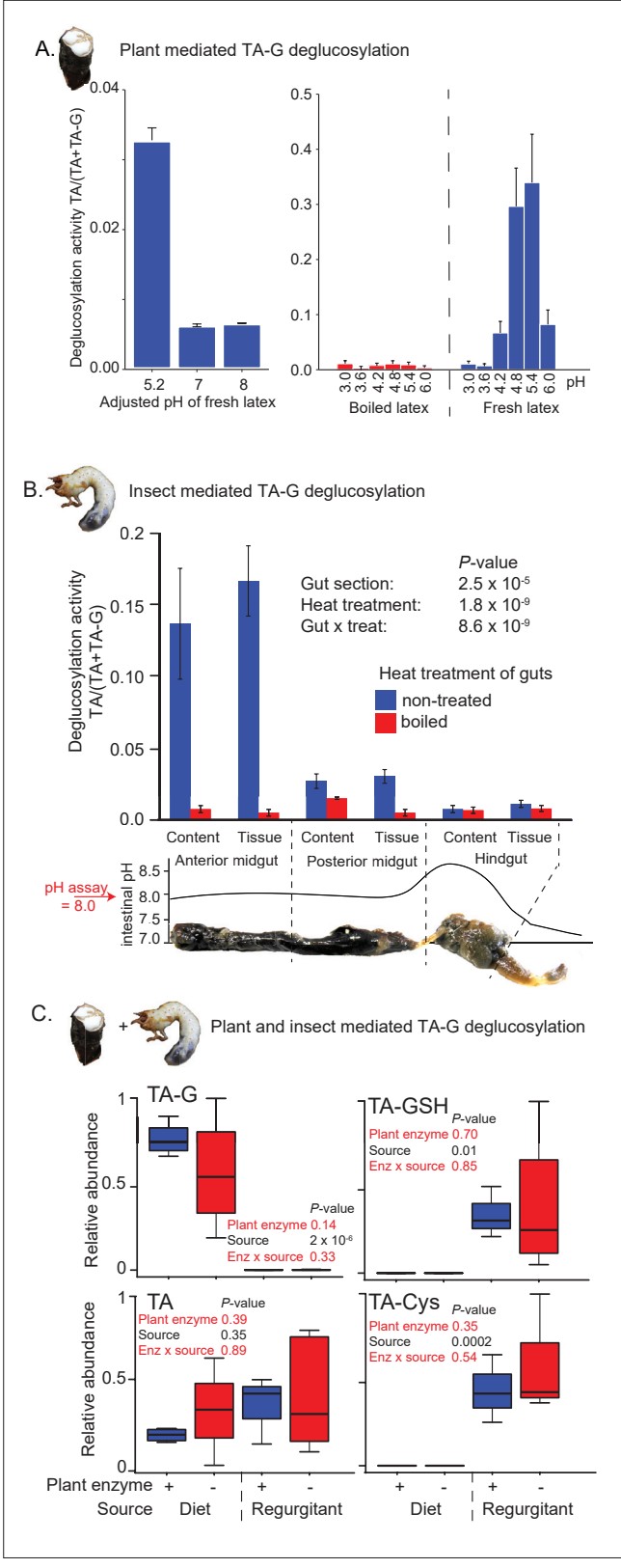

**Figure 2.** Insect rather than plant enzymes deglucosylate TA-G. (**A**) Left and right panels: plant-mediated enzymatic deglucosylation of TA-G at pH 3–8. *Taraxacum officinale* latex was collected from wounded roots and incubated in buffers adjusted to different pH values. N = 3. (**B**) Deglucosylation activity of untreated and boiled extracts of *Melolontha melolontha* gut content and gut tissue incubated at pH 8.0 with boiled latex extracts.

*Figure 2 continued on next page*

*Figure 2 continued*

The p-values of a two-way analysis of variance (ANOVA) are shown. N = 6. Error bars = SEM. The intestinal pH of *M. melolontha* is shown for comparative purposes (data from *Egert et al., 2005*). For in vitro *M. melolontha* glucosidase inhibition assays, refer to *Figure 2—figure supplement 1* - 2. (**C**) Relative abundance of TA-G and its metabolites in the diet and regurgitant of larvae fed with carrot slices coated with either intact (+) or heat-deactivated (-) *T. officinale* latex. Heat deactivation of latex did not significantly affect the deglucosylation of TA-G in *M. melolontha*. p-values refer to two-way ANOVAs. N = 4. TA-G: taraxinic acid $\beta$-D-glucopyranosyl ester; TA = taraxinic acid; GSH = glutathione; Cys = cysteine. Peak area was normalized across all treatments based on the maximal value of each metabolite. Raw data are available in *Figure 2—source data 1*.

The online version of this article includes the following figure supplement(s) for figure 2:

**Source data 1.** Source data of main and supplementary figures of *Figure 2*.

**Figure supplement 1.** TA-G deglucosylation activity (TA/(TA + TA -G)) of *Melolontha melolontha* anterior midgut samples in the presence of either the carboxylesterase inhibitor bis(p-nitrophenyl)phosphate or the α- and $\beta$-glucosidase inhibitor castanospermin.

**Figure supplement 2.** TA-G deglucosylation activity (TA/(TA + TA G)) of *Melolontha melolontha* anterior midgut samples in the presence of acarbose, an α-glucosidase-specific inhibitor, or castanospermine, an α- and $\beta$-glucosidase inhibitor.

*supplement 4*). Apart from TA-G, Mm_bGlc17 also deglyosylated benzoxazinoids, salicin, and cellobiose. These data suggest that Mm_bGlc17 and up to four other gut-expressed β-glucosidases may play a role in TA-G metabolism in *M. melolontha*.

## The *M. melolontha* b-glucosidase Mm_bGlc17 hydrolyzes TA-G in vivo

To test whether *M. melolontha* β-glucosidases contribute to TA-G deglucosylation, we silenced two β-glucosidases with TA-G deglucosylation activity, Mm_bGlc16 and Mm_bGlc17, as well as one β-glucosidase without TA-G activity, Mm_bGlc18, by injecting double-stranded RNA (dsRNA) targeting a 500 bp fragment of each gene into the second segment of anesthetized *M. melolontha* larvae (*Figure 4—figure supplement 1*). After 5 days, a stable and specific reduction of the target mRNAs had occurred (*Figure 4—figure supplements 2–3*). TA-G deglucosylation was reduced by 75 % in gut extracts of larvae that were silenced in *Mm_bGlc17* (*Figure 4A*). Silencing of *Mm_bGlc16* and *Mm_bGlu18* did not significantly reduce TA-G deglucosylation activity compared to green fluorescent protein (GFP) controls (*Figure 4A*). These results confirm that *M. melolontha*-derived β-glucosidases hydrolyze TA-G and demonstrate that Mm_Glc17 accounts for most of the TA-G deglucosylation in vivo.

## Mm_bGlc17 benefits *M. melolontha* growth on TA-G-containing plants

To test whether Mm_bGlc17 modulates the impact of TA-G on larval performance, *Mm_bGlc17*-silenced and *GFP*-control larvae were allowed to feed on either TA-G-producing wild-type or TA-G-deficient transgenic dandelions. The interaction of *Mm_bGlc17* silencing and plant genotype significantly affected larval growth (*Figure 4B*; p(*Mm_bGlc17* x TA-G) = 0.009, two-way analysis of variance (ANOVA)). On TA-G-containing plants, *Mm_bGlc17* silencing reduced larval growth, with *GFP*-control larvae gaining 4.5 % body weight and *Mm_bGlc17*-silenced larvae losing 1.4 % body weight (*Figure 4B*; p = 0.009, Student's t-test). By contrast, on TA-G-deficient plants, *Mm_bGlc17* silencing did not affect larval weight gain (p = 0.19, Student's t-test). *GFP*-control *M. melolontha* larvae had higher growth on TA-G-containing than TA-G-lacking plants (p = 0.035, Student's t-test; *Figure 4*, *Figure 4—figure supplement 4*), while the reversed pattern was found in tendency for *Mm_bGlc17*-silenced larvae (p = 0.099, Student's t-test; *Figure 4—figure supplement 4*). The experiment was repeated twice with similar results (*Figure 4—figure supplement 5*). As Mm_bGlc17 benefited larval growth in the presence of TA-G, we investigated whether the expression of this gene is induced by TA-G. *Mm_bGlc17* gene expression increased by 95 % on TA-G-containing compared to TA-G-lacking plants (*Figure 4C*; p = 0.04, Kruskal-Wallis rank sum test). Taken together, these data show that *Mm_bGlc17* expression is induced by TA-G and increases larval performance in the presence of TA-G.

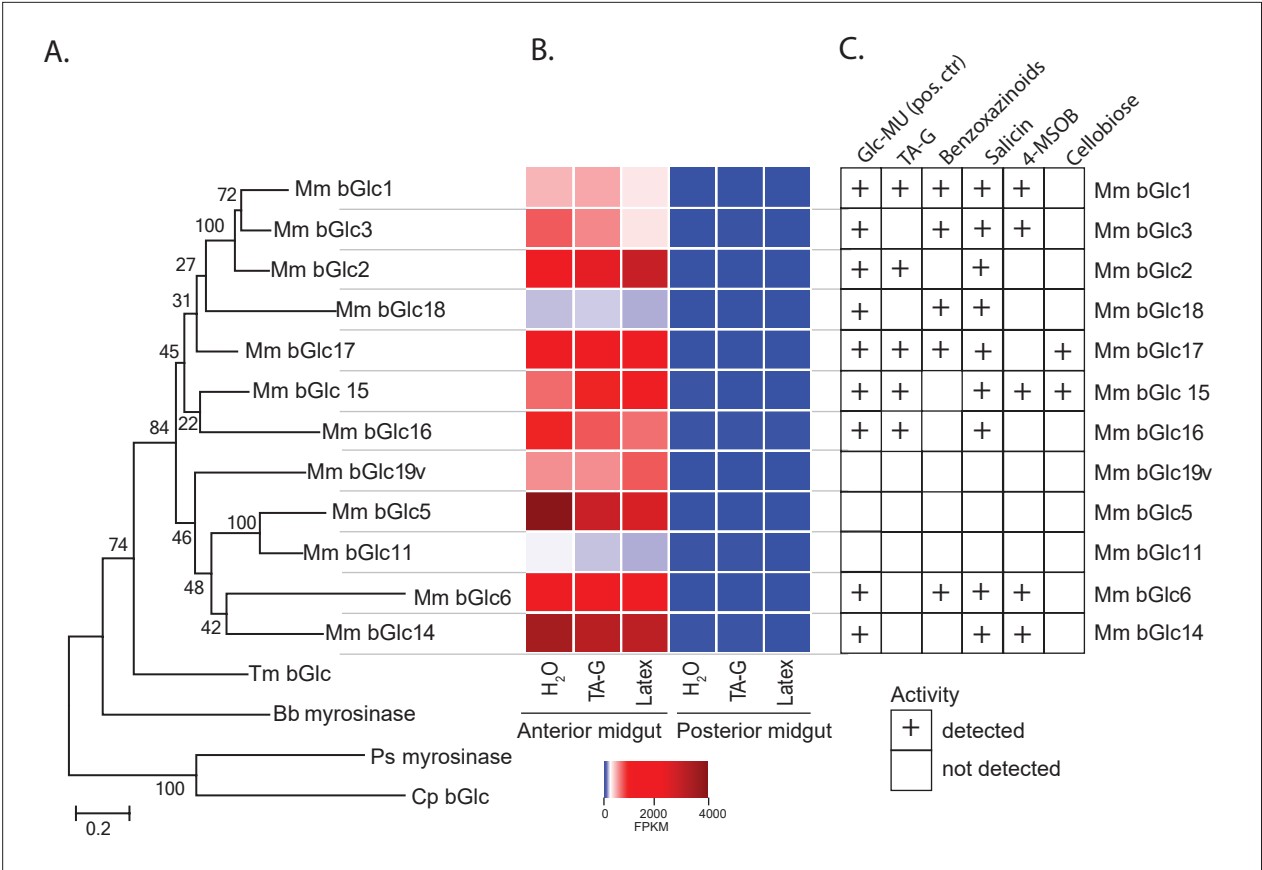

**Figure 3.** *Melolontha melolontha* midgut $\beta$-glucosidases hydrolyze TA-G and other plant defensive glycosides. (**A**) Phylogeny of newly identified *Melolontha melolontha* $\beta$-glucosidases and previously reported $\beta$-glucosidases of *Tenebrio molitor* (Tm bGlc, AF312017.1) and *Chrysomela populi* (Cp bGlc, KP068701.1), and myrosinases (thioglucosidases) of *Phyllotreta striolata* (Ps myrosinase, KF377833.1) and *Brevicoryne brassicae* (Bb myrosinase, AF203780.1) based on amino acid similarities using maximum likelihood method. Bootstrap values (N = 1000) are shown next to each node. Amino acid sequence alignments of the $\beta$-glucosidases are shown in *Figure 3—figure supplement 1*. (**B**) Heat map of average (n = 3) gene expression levels of *M. melolontha* $\beta$-glucosidases in the anterior and posterior midgut of larvae feeding on diets supplemented with water, taraxinic acid $\beta$-D-glucopyranosyl ester (TA-G), or *Taraxacum officinale* latex-containing diet. FPKM = fragments per kilobase of transcript per million mapped reads. (**C**) Activity of heterologously expressed *M. melolontha* $\beta$-glucosidases with TA-G, a mixture of maize benzoxazinoids, the salicinoid salicin, 4-methylsulfinylbutyl glucosinolate (4-MSOB), cellobiose, and the fluorogenic substrate 4-methylumbelliferyl-$\beta$-D-glucopyranoside (Glc-MU). Glucosidase activities of three consecutive assays with excreted proteins from insect High Five cells were measured. Negative controls (buffer, non-transfected wild-type cells, and cells transfected with green fluorescent protein) did not hydrolyze any defense metabolite. Results from the individual assays are shown in *Figure 3—figure supplement 2*. For deglycosylation of these compounds by *M. melolontha* gut protein crude extracts, refer to *Figure 3—figure supplement 3*. Deglycosylation assays with recombinant Mm_bGlc17 yielded highest aglycone formation; *Figure 3—figure supplement 4*. Raw data are available in *Figure 3—source data 1*.

The online version of this article includes the following figure supplement(s) for figure 3:

**Source data 1.** Source data of main and supplementary figures of *Figure 3*.

**Figure supplement 1.** Amino acid sequence alignment of $\beta$-glucosidases of *Melolontha melolontha* and other insect species.

**Figure supplement 2.** Activity of heterologously expressed *Melolontha melolontha* $\beta$-glucosidases and negative controls (GFP = green fluorescent protein; WT = non-transfected wild type; buffer) toward plant defensive glycosides, cellobiose, and the standard substrate 4-methylumbelliferyl-$\beta$-D-glucopyranoside (Glc-MU), which fluoresces upon deglucosylation.

**Figure supplement 3.** Deglucosylation of defensive glycosides by boiled and non-boiled *Melolontha melolontha* anterior midgut extracts in vitro.

**Figure supplement 4.** Taraxinic acid aglycone formation of the heterologously expressed *Melolontha melolontha* $\beta$-glucosidases and negative controls (GFP = green fluorescent protein; WT = non-transfected wild type; buffer) of two deglucosylation assays.

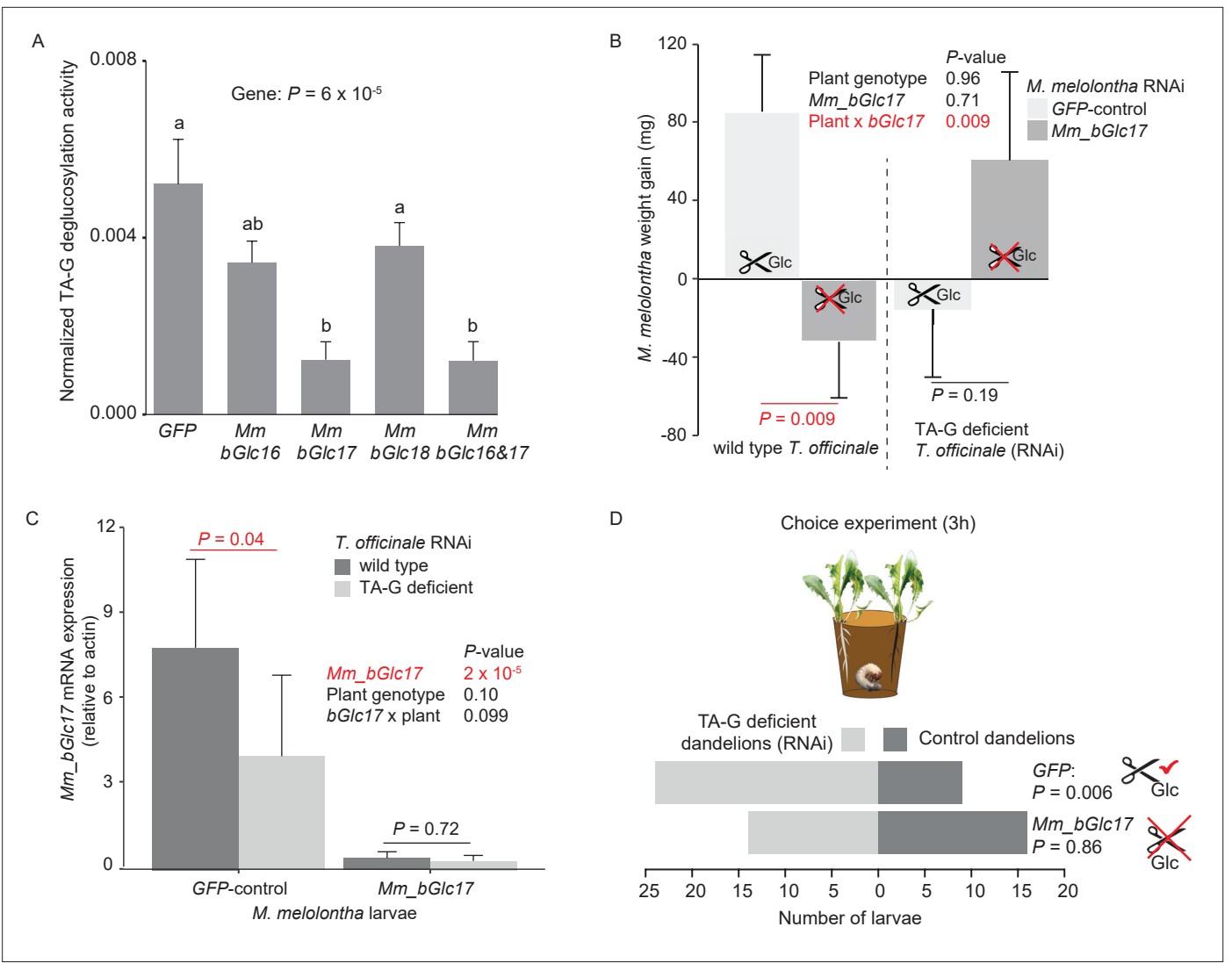

**Figure 4.** Silencing of *Mm_bGlc17* reduces TA-G deglucosylation and modifies the impact of TA-G on larval growth and host plant choice. (**A**) Taraxinic acid $\beta$-D-glucopyranosyl ester (TA-G) deglucosylation activity (TA/(TA + TA G)) of gut extracts from *Melolontha melolontha* larvae in which different $\beta$-glucosidases were silenced through RNA interference (RNAi), resulting in stable and specific silencing of the individual glucosidases, *Figure 4—figure supplements 2–3*. Silencing of *Mm_bGlc17* significantly reduced hydrolysis of TA-G by gut extracts. A green fluorescent protein-derived double-stranded RNA (*GFP* dsRNA) construct was used as a negative control. *Mm_bGlc16&17*-treated larvae received a 50:50 (v/v) mixture of both dsRNA species. Deglucosylation activity was normalized to that of boiled control samples to correct for the background of non-enzymatic hydrolysis. N = 9–10. p-value of a one-way analysis of variance (ANOVA) is shown. Different letters indicate a significant difference according to Tukey's honest significance test. Error bars = SEM. (**B**) Weight gain of *Mm_bGlc17*-silenced and *GFP*-control *M. melolontha* larvae growing on transgenic TA-G-deficient or control *Taraxacum officinale* lines. N = 11–15. p-values refer to a two-way ANOVA and Student's t-tests. Error bars = SEM. For comparing growth of *GFP*- and *Mm_bGlc17*-silenced larvae between TA-G-deficient and control lines, refer to *Figure 4—figure supplement 4*. The experiment was repeated once with similar results; *Figure 4—figure supplement 5*. (**C**) Gene expression (relative to actin) of *Mm_bGlc17*-silenced and *GFP*-control *M. melolontha* larvae feeding on transgenic TA-G-deficient or control *T. officinale* lines. N = 12–14. p-values refer to a two-way ANOVA (log-transformed data) and Kruskal-Wallis rank sum tests (non-transformed values). (**D**) Choice of *Mm_bGlc17*-silenced and *GFP*-control larvae between transgenic TA-G-deficient and control *T. officinale* lines. Silencing of *Mm_bGlc17* abolished the choice of control larvae for TA-G-deficient lines. p-values refer to binomial tests. Choice was stable over time; see *Figure 4—figure supplement 6*. Raw data are available in *Figure 4—source data 1*.

The online version of this article includes the following figure supplement(s) for figure 4:

**Source data 1.** Source data of main and supplementary figures of *Figure 4*.

**Figure supplement 1.** Injection of dsRNA by a sterile syringe between the second and third segment of *Melolontha melolontha*.

**Figure supplement 2.** *Tubulin* mRNA expression in *Melolontha melolontha* larvae treated with 2.5 µg or 0.25 µg *GFP* or *tubulin* dsRNA per g larval mass 2, 5, and 10 days after injection (N = 3).

*Figure 4 continued on next page*

*Figure 4 continued*

**Figure supplement 3.** Gene expression (relative to actin) of *Mm_bGlc16*, *Mm_bGlc17*, and *Mm_bGlc18* in *Mm_bGlc17* dsRNA-injected larvae 2 days after dsRNA application.

**Figure supplement 4.** Weight gain of *Mm_bGlc17*-silenced and *GFP*-control *Melolontha melolontha* larvae growing on transgenic TA-G-deficient or control *Taraxacum officinale* lines (N = 11–15).

**Figure supplement 5.** Repetition of the experiment on weight gain of *Mm_bGlc17*-silenced and *GFP*-control *Melolontha melolontha* larvae growing on transgenic TA-G-deficient or control *Taraxacum officinale* lines.

**Figure supplement 6.** Choice of *Melolontha melolontha* larvae that were treated with either *GFP* (**A**) or *Mm_bGlc17* (**B**) dsRNA between TA-G-deficient transgenic and wild-type *Taraxacum officinale* plants 1–4 hr after the start of the experiment.

## *Mm_bGlc17* expression is required for the deterrent effect of TA-G toward *M. melolontha*

As TA-G in *T. officinale* latex was previously found to deter *M. melolontha* larvae (*Pollard, 1992*), we tested whether TA-G hydrolysis influences the deterrent properties of TA-G. *Mm_bGlc17*-silenced and *GFP*-control larvae were allowed to choose between TA-G-producing wild-type and TA-G-deficient transgenic dandelions. *GFP*-silenced control larvae were deterred by TA-G, with over 60 % of the larvae feeding on TA-G-deficient plants and 30 -% on the wild-type (*Figure 4D*; p(3h) = 0.006, binomial test). By contrast, *Mm_bGlc17*-silenced larvae did not show any preference for TA-G-deficient over TA-G-producing wild-type plants: 44 % of the larvae fed on wild-type plants, while 42 % fed on TA-G-deficient plants (*Figure 4D*; p(3h) = 0.86, binomial test). Both patterns were constant over time (*Figure 4—figure supplement 6*). *Mm_bGlc17* silencing did not significantly affect the total percentage of larvae that made a choice (86% Mm_bGlc17 vs 91 % GFP). These results demonstrate that *Mm_bGlc17* expression is required for the deterrent effect of TA-G toward *M. melolontha*.

## Discussion

Herbivore enzymes are well known to modify plant defense metabolites, but only few studies provided clear evidence that these modifications feed back on herbivore performance and fitness. Furthermore, the effects of plant defense metabolizations on herbivore host plant choice are not understood. Here, we show that a herbivore β-glucosidase deglucosylates a plant secondary metabolite, which modifies both its toxic and deterrent properties and thereby determines the interaction between a plant and its major root-feeding natural enemy.

Metabolization of plant defense metabolites is considered central for the ability of species to overcome chemical defenses of their host plants (*Heckel, 2014*), and recent papers have established direct molecular evidence for this concept (*Sun et al., 2019*; *Sun et al., 2020*; *Poreddy et al., 2015*). A major metabolization product of TA-G is TA-Cys, with about 25 % of the ingested TA-G accumulating in this form. Based on our current knowledge of the GSH pathway in insects (*Schramm et al., 2012*), it is likely that TA-G is deglucosylated prior conjugation to GSH and subsequently sequentially cleaved to TA-Cys by peptidases. The first step of this metabolization pathway, the conjugation of GSH to TA, may occur spontaneously and/or via GSH glucosyl-transferases. When we incubated TA and TA-G with high concentrations of GSH and Cys in vitro, several isomers of the conjugates formed that were not detected inside the *M. melolontha* gut, suggesting that enzymatic rather than spontaneous conjugation of GSH to TA prevails inside the larva. Interestingly, TA-Cys mostly accumulated in the anterior midgut, and only neglectable amounts of TA-Cys were excreted by the larvae, a pattern that was stable over long-term feeding of *M. melolontha* on *T. officinale*. Consequently, larvae must further metabolize TA-Cys to yet unknown products, and either store or excrete these compounds. Future experiments with radioisotope-labeled TA-G may shed light into the ultimate fate of TA inside the *M. melolontha* larva, and may help to assess whether *M. melolontha* sequesters TA and uses the compound for its own prupose.

As the transformation of defense metabolites by insect enyzmes occurs in the gut, metabolization products are considered unlikely to be tasted via frontal sensory structures of insect herbivores. It is thus commonly assumed that there is no direct impact of this process on herbivore behavior (*Pentzold et al., 2014*; *Simon et al., 2015*). By contrast, transformation of defense metabolites by plant

enzymes that are activated by tissue disruption is well accepted to have a strong behavioral impact on insect herbivores, which is in line with the rapid and early formation of plant defense catabolites (*Glauser et al., 2011*; *Krothapalli et al., 2013*; *de Vos et al., 2008*; *Zhang et al., 2006*; *Mumm et al., 2008*). Here, we have found that the insect β-glucosidase Mm_bGlc17, which deglucosylates a defensive sesquiterpene lactone (TA-G) in the insect gut, is also required to elicit the deterrent effect of this metabolite. Our early work on TA-G showed that, in a community context, the capacity of dandelions to produce the glucosylated sesquiterpene lactone reduces *M. melolontha* attack and its negative effect on plant growth and fitness (*Huber et al., 2016b*), resulting in the selection of high TA-G genotypes under high *M. melolontha* pressure (*Huber et al., 2016a*). As these effects are likely the result of the deterrent, rather than the toxic properties of TA-G, they are likely also directly dependent on the presence of Mm_bGlc17. Thus, the metabolism of *M. melolontha* may not only drive the feeding preferences of the herbivore, but also the ecology and evolution of dandelions in their natural habitat. Insect-detoxifying enzymes may thus shape plant defense evolution not only by reducing the toxicity of defense compounds but also by modulating herbivore host plant choice.

Many plant defensive metabolites are glycosides, which are typically non-toxic themselves but are deglucosylated upon herbivore damage, forming toxic products. Both plant- and herbivore-derived β-glucosidases can mediate deglycosylation in the insect gut, but their relative contribution is often unclear (*Pentzold et al., 2014*; *Desroches et al., 1997*; *Lindroth, 1988*; *Pankoke et al., 2012*). Here we provide several parallel lines of evidence to demonstrate that the deglucosylation of TA-G, a glucosylated secondary metabolite in the latex of *T. officinale*, depends primarily on β-glucosidases from *M. melolontha* rather than on plant enzymes. First, *T. officinale* TA-G hydrolase activity has an acidic pH optimum (4.8–5.4), and the activity is very low at the alkaline pH (8.0) found in the gut of *M. melolontha*. Second, TA-G is deglucosylated by *M. melolontha* gut extracts in the absence of plant material. Third, the presence of TA-G-hydrolyzing latex proteins in TA-G-containing diet does not result in higher amounts of TA or TA conjugates inside *M. melolontha* compared to the diet with heat-deactivated latex proteins. Fourth, *M. melolontha* expresses several β-glucosidases with TA-G-hydrolyzing activity as demonstrated in in vitro assays. Fifth, silencing the *M. melolontha* TA-G β-glucosidase Mm_bGlc17 reduces TA-G deglucosylation activity in larval gut extracts and abolishes the avoidance behavior of *M. melolontha* toward TA-G-containing plants. Together, these results demonstrate that insect rather than plant β-glucosidases hydrolyze ingested TA-G in *M. melolontha*.

A large number of plant glycosides are protoxins that are activated by deglycosylation including glucosinolates, benzoxazinoids, salicinoids, alkaloid glycosides, cyanogenic glycosides, and iridoid glycosides (*Mithöfer and Boland, 2012*; *Pentzold et al., 2014*; *Wittstock and Gershenzon, 2002*). But, until now nothing was known about whether sesquiterpene lactone glycosides are also protoxins. Sesquiterpene lactone aglycones are much more potent than their corresponding glycosides in pharmacological studies of cytotoxicity and anti-cancer activity (*Choi et al., 2002*; *Seto et al., 1988*). However, the consequences of sesquiterpene lactone deglycosylation for herbivore behavior and performance have not been previously investigated (*Huber et al., 2015*; *Sessa et al., 2000*; *Graziani et al., 2015*). Our experiments show that deglucosylation of TA-G is associated with an increase rather than a decrease in larval growth on TA-G-producing plants. This suggests that the cleavage of TA-G to TA reduces rather than enhances the toxicity of this sesquiterpene lactone. Several explanations for this phenomenon are possible. First, GSH may be more rapidly conjugated by TA than TA-G, and thus deglucosylation is a step toward detoxification. Second, if the target site of TA-G lies in a hydrophilic compartment (such as the gut lumen), deglucosylation may block its activity. Third, the glucose liberated by TA-G deglucosylation may enhance the nutritional quality of dandelion roots for the larvae. When we compared dandelion roots exposed to different native grassland species in previous studies, we found both positive and negative correlations between root glucose levels and larval growth (*Huang et al., 2019*; *Huang et al., 2018*), suggesting a high degree of context dependency. In summary, our results provide evidence that deglycosylation of plant defenses may reduce negative impacts on herbivores. Deglycosylation of a diterpene glycoside of *N. attenuata* was also found to reduce its toxicity, but in this case, the product still contained two other glycoside moieties and thus differs little from its substrate in terms of polarity compared to the differences between TA and TA-G (*Poreddy et al., 2015*).

While Mm_bGlc17 improves larval performance on TA-G-producing plants, the enzyme is also required for *M. melolontha* larvae to avoid TA-G. We propose two mechanisms that may be responsible

for these counterintuitive results. First, the recognition of TA-G through deglucosylation may guide the *M. melolontha* larva to feeding sites that are most suitable for fast larval growth, independently of the toxicity of TA-G. Exploitation of plant secondary metabolites and sugars to locate nutritious tissue has been reported, for instance, for the specialist root herbivore *Diabrotica virgifera virgifera* feeding on maize roots (*Robert et al., 2012*; *Hu et al., 2018*; *Machado et al., 2021*). *Melolontha melolontha* larvae preferentially feed on side roots of dandelions, which contain lower TA-G and higher soluble protein levels than main roots and also may be more nutritious as they are actively growing (*Huber et al., 2016b*). Thus, the larvae may not be avoiding TA-G because of its toxicity, but because avoiding high TA-G levels guides them to nutritious roots, with the avoidance behavior being facilitated by *Mm_bGlc17*.

A second explanation for the observed patterns may be that herbivore growth by itself gives an incomplete picture regarding the costs of TA-G consumption and metabolism. It has been shown, for instance, that plant secondary metabolites can enhance larval weight gain, but at the same time increase larval mortality, suggesting that growth is not always beneficial (*Veyrat et al., 2016*; *Erb, 2018*). Furthermore, TA may change the susceptibility of the larvae to parasites and pathogens, as has been shown for the plant volatile indole in maize (*Ye et al., 2018*). In addition, the hydrolysis of TA-G may deplete the level of cysteine inside the larva through conjugation of TA to glutathione, as has been observed in lepidopteran larvae that conjugate food-derived isothiocyanates to GSH (*Jeschke et al., 2016*). The negative consequences of Cys depletion on larval performance may only be observed under stressful conditions, for instance, under nutrient limitations or in the presence of other toxic allelochemicals that require detoxification through GSH conjugation. Thus, it is possible that under natural conditions, Mm_bGlc17-dependent cleavage of TA-G reduces rather than enhances *M. melolontha* fitness, which may explain Mm_bGlc17-mediated TA-G avoidance. The gene may, nevertheless, be maintained in the insect genome if Mm_bGlc17 is important for the larva to acquire nutrients. In such a scenario, dandelions would exploit a promiscuous β-glucosidase (Mm_bGlc17) whose evolution is constrained by its primary functions in nutrient acquisition.

If the evolution of Mm_bGlc17 is constrained by its primary function, such constraints may be alleviated by down-regulating the expression of the gene under harmful conditions, as has been observed for other insect β-glucosidases in the presence of glycosydic protoxins (*Pentzold et al., 2014*). In our study, while the initial RNA-sequencing (RNAseq) analysis did not detect differences in *Mm_bGlc17* expression in the presence and absence of TA-G, our follow-up quantitative PCR (qPCR) analysis on *M. melolontha* larvae feeding on transgenic plants showed that *Mm_bGlc17* is up-regulated in the presence of TA-G. Further experiments are required to determine whether this discrepancy is due to different methods and sample sizes or due to differences in TA-G concentration, the presence of a plant matrix, or the genotypic background of the *M. melolontha* larvae. Despite these uncertainties, the observed up-regulation of *Mm_bGlc17* expression in the presence of TA-G is compatible with a beneficial role of this enzyme for *M. melolontha* feeding on TA-G-producing dandelions. A more detailed understanding of the role of Mm_bGlc17 and TA-G under natural conditions and over the full 3 -year life cycle of *M. melolontha* would help to shed light on whether the expression of *Mm_bGlc17* is indeed beneficial for the larvae.

While Mm-bGlc17 is required for the feeding deterrence of TA-G, the underlying physiological mechanisms that lead to TA-G avoidance are unclear. On the one hand, insect feeding preference may be triggered by the presence of the aglycone TA or TA conjugates inside the gut. These metabolites may bind to specific receptors that face the gut lumen or are located inside the gut membrane, and thereby alter herbivore host plant choice. Additionally, reduction of GSH or Cys levels through conjugation to TA may be perceived by the larvae and thereby alter herbivore feeding preference, although this scenario is less likely as GSH or Cys depletion likely requires longer time than the immediately observed feeding responses of naïve *M. melolontha* larvae. On the other hand, TA or its conjugates may be present at low concentrations inside the *M. melolontha* mouth, for instance, via regurgitation or through direct formation due to the potential presence of Mm_bGlc17 in the saliva, and may thus be detected by gustatory receptors that modulate herbivore host plant choice. We found that forced regurgitants of *M. melolontha* possess TA-G-hydrolyzing activity; however, it is unclear whether forced regurgitants are informative to infer normal feeding processes of the larvae. While Mm_bGlc17 is highly abundant in the gut, it is unclear whether Mm_bGlc17 is also present in the saliva. Analyzing the expression profile of *Mm_bGlc17* across different organs of the *M. melolontha* larva, as well as

performing neurosensory experiments with orally administered TA, would help to shed light onto the exact mechanisms underlying TA feeding deterrence.

Interestingly, besides TA-G, Mm_bGlc17 deglycosylates other substrates, including cellobiose and salicinoid and benzoxazinoid defense compounds. The ability of this enzyme to hydrolyze benzoxazinoids seems counterintuitive from the insect's perspective since benzoxazinoid hydrolysis increases both feeding deterrence as well as toxicity (*Glauser et al., 2011*; *de Vos et al., 2008*; *Zhang et al., 2006*; *Mumm et al., 2008*; *Wittstock et al., 2003*), raising the possibility that some plants can co-opt insect enzymes to activate their own defenses. On the other hand, insects are known to have evolved some resistance to plant glycosidic protoxins by inhibiting the activating glycosidases of plants and down-regulating their own activating glycosidases (*Pentzold et al., 2014*; *Desroches et al., 1997*; *Lindroth, 1988*). The fact that Mm_bGlc17 catalyzes the hydrolysis of a range of glucosides plus the glucose ester TA-G is also unusual. There are only a few previous reports of enzymes with this versatility (*Nakano et al., 1998*; *Okamoto et al., 2000*).

The ability of Mm_bGlc17 to mediate hydrolysis of cellobiose, a disaccharide derived from cellulose, suggests its evolutionary origin as a digestive enzyme that was later recruited for processing plant defenses. The relatively large number of β-glucosidases in many insect herbivores (*Pentzold et al., 2014*; *Poreddy et al., 2015*; *Beran et al., 2014*) and their species-specific phylogenetic clustering (*Beran et al., 2014*) indicate that in addition to contributing to the digestion of cell wall carbohydrates—which are mostly shared among plant species—many β-glucosidases also act on a variety of specialized metabolites, such as plant defense compounds. Thus, plant defenses may play an underestimated role in the evolution of β-glucosidases in insect herbivores. Other herbivore digestive enzymes may also interact with plant defenses, leading to changes in herbivore performance and behavior, which likely modulate the ecology and evolution of plants and their consumers.

# Materials and methods

**Key resources table**

| Reagent type (species) or resource | Designation | Source or reference | Identifiers | Additional information |
|---|---|---|---|---|
| Gene (*Melolontha melolontha*) | *Mm_bGlc17* | This paper | | See *Supplementary file 1* |
| Genetic reagent (*Taraxacum officinale*) | TA-G-deficient plants | Doi:10.1371/journal.pbio.1002332 | A34-RNAi-1 | |
| Genetic reagent (*M. melolontha*) | *Mm_bGlc17* silenced | This paper | | See 'Materials and methods' ('TA-G deglucosylation activity in RNAi-silenced *M. melolontha* larvae') |
| Cell line (*Trichoplusia ni*) | High Five Cells | Life Technologies, Carlsbad, CA, USA | | |
| Transfected construct (*T. ni*) | pIB/V5-His TOPO | Life Technologies, Carlsbad, CA, USA | | |
| Biological sample (*M. melolontha*) | *M. melolontha* | This paper | | Collection of different natural populations |
| Biological sample (*T. officianle*) | *T. officinale* | Doi: | | Different natural genotypes |
| Recombinant DNA reagent | Plasmid pGJ 2648 | Other | | Supplied by Dr. Christian Schulze-Gronover, Fraunhofer Institute for Molecular Biology and Applied Ecology |
| Recombinant DNA reagent | pCR2.1-TOPO plasmids | Life Technologies | | |
| Recombinant DNA reagent | cDNAs | This paper | | See *Supplementary file 3* |
| Sequence-based reagent | Primers | This paper | | See *Supplementary file 2* |

*Continued on next page*

*Continued*

| Reagent type (species) or resource | Designation | Source or reference | Identifiers | Additional information |
|---|---|---|---|---|
| Sequence-based reagent | KAPA SYBR FAST qPCR Master Mix | Kapa Biosystems | | |
| Commercial assay or kit | innuPREP RNA Mini Kit | Analytik Jena, Jena, Germany | | |
| Commercial assay or kit | RNeasy Plant Mini Kit; RNeasy Lipid Tissue Mini Kit | Qiagen | | |
| Commercial assay or kit | SMARTer RACE cDNA Amplification Kit | Clontech, Mountain View, CA, USA | | |
| Commercial assay or kit | FuGeneHD-Kit | Promega, Madison, WI, USA | | |
| Commercial assay or kit | GeneJET Gel Extraction Kit; DreamTaq DNA Polymerase; MEGAscript RNAi Kit | Thermo Fisher Scientific, Waltham, MA, USA | | |
| Commercial assay or kit | KAPA SYBR FAST qPCR Kit Optimized for LightCycler 480 | Kapa Biosystems, Wilmington, MA, USA | | |
| Chemical compound, drug | Bis(p-nitrophenyl)phosphate; castanospermine; acarbose | Sigma Aldrich | | |
| Chemical compound, drug | Blasticidin; Express Five culture medium | Life Technologies | | |
| Chemical compound, drug | x Protease Inhibitor HP Mix | SERVA Electrophoresis, Heidelberg, Germany | | |
| Chemical compound, drug | TA-G; TA; BXDs; 4-MSOB | This paper | | See 'Materials and methods' ('Enzymatic assays of recombinant proteins' and 'Synthesis of TA-G metabolite standards') |
| Chemical compound, drug | Salicin | Alfa Aeser | | |
| Chemical compound, drug | Cellobiose | Fluka | | |
| Chemical compound, drug | Glc-MU | Sigma Aldrich | | |

## Plant material

*T. officinale* plants used for extraction of latex and TA-G were grown in 0.7–1.2 mm sand and watered with 0.01–0.05% fertilizer with N-P-K of 15-10-15 (Ferty 3, Raselina, Czech Republic) in a climate chamber operating under the following conditions: 16 hr light/8 hr dark; light supplied by a sodium lamp (EYE Sunlux Ace NH360FLX, Uxbridge, UK); light intensity at plant height: 58 µmol m$^2$ s$^{-1}$; temperature: day 22 °C; night 20 °C; humidity: day 55%, night 65 %. Depending on the availability, 3- to 5-month-old wild-type plants of the European A34, 6.56, or 8.13 accession were used unless otherwise indicated (*Verhoeven et al., 2010*). Plants used for the choice experiments were germinated on seedling substrate and transplanted into individual pots filled with potting soil (five parts landerde, four parts peat, and one part sand) after 2–3 weeks and grown in a climate chamber operating under the following conditions: 16 hr light/8 hr dark, light supplied by arrays of Radium Bonalux Super NL 39 W/840 white lamps; light intensity at plant height: 250 µmol m$^2$ s$^{-1}$; temperature: day 22 °C; night 18 °C; humidity 65 %. Plants used for the performance experiments were germinated on seedling substrate, transplanted to individual pots filled with a homogenized mixture of 2/3 seedling substrate (Klasmann-Deilmann, Switzerland) and 1/3 landerde (Ricoter, Switzerland) and cultivated in a greenhouse operating under the following conditions: 50–70% relative humidity, 16/8 hr light/dark cycle, and 24 °C at day and 18 °C at night, without extrernal light source. The TA-G-deficient line RNAi-1 and the control line RNAi-15 were used for these experiments (*Huber et al., 2016b*).

## Insects

*M. melolontha* larvae were collected from meadows in Switzerland and Germany. Larvae were reared individually in 200 ml plastic beakers filled with a mix of potting soil and grated carrots in a climate chamber operating under the following conditions: 12 hr day, 12 hr night; temperature: day 13 °C, night 11 °C; humidity: 70 %; lighting: none, except for the RNAi experiment, for which the day and night temperature was 4 °C during rearing. All experiments were performed in the dark with larvae in the third larval instar.

## Cell lines

For heterologous experession, *T. ni*-derived cells (High Five cells) were purchased from Life Technologies (Carlsbad, CA, USA) and immediately used for the experiment. The cell lines were tested negatively for mycoplasma infection prior delivery.

## Statistical analysis

All statistical analyses were performed in R version 3.1.1 (*R Development Core Team, 2014*). Pairwise comparisons were performed with the Agricolae package (*de Mendiburu, 2014*). Results were displayed with gplots, ggplot2, and RColorBrewer (*Wickham, 2009*; *Warnes et al., 2016*; *Neuwirth, 2014*). Differential gene expression was analyzed using DeSeq2 and edgeR (*Robinson et al., 2010*; *Love et al., 2014*). Details on the statistical procedure are given in the individual sections. Sample sizes were estimated based on previous experience with the study system. *M. melolontha* larvae were allocated to treatment groups using restricted randomization to achieve equal sample sizes among groups.

## Isolation and identification of TA-G metabolites in *M. melolontha* larvae

In order to test whether TA-G is deglucosylated during digestion in *M. melolontha*, we screened for TA-G, TA, and other TA-G metabolites in larvae that fed on diets supplemented with either latex or water. 10 *M. melolontha* larvae were starved for 10 days at room temperature before offering them approximately 0.35 $cm^3$ boiled carrot slices that were coated with either main root latex or water. Larvae were allowed to feed for 4 hr inside 180 ml plastic beakers covered with a moist tissue paper, after which the frass and regurgitant were collected in 1 ml methanol. Regurgitant was collected by gentle prodding of the larvae. Left-over food was frozen at –80 °C until extraction. The larvae were cooled for 10 min at –20 °C and subsequently dissected on ice to remove the anterior midgut, posterior midgut, hindgut, and hemolymph, which were collected in 1 ml methanol. All larval samples were homogenized by vigorously shaking with two to three metal beads for 4 min in a paint shaker (Fluid Management, Wheeling, IL, USA), centrifuged at 4 °C for 10 min at 17,000 ×*g*, and the supernatant stored at –20 °C until analysis. Left-over food was ground in liquid nitrogen to a fine powder of which 100 mg was extracted with 1 ml methanol by vortexing for 30 s. The samples were subsequently centrifuged at room temperature for 10 min at 17,000 ×*g* and the supernatant was stored at –20 °C until analysis. Methanol samples were analyzed on a high-pressure liquid chromatograph (HPLC 1100 series equipment; Agilent Technologies, Santa Clara, CA, USA), coupled to a photodiode array detector (G1315A DAD; Agilent Technologies) and an Esquire 6,000 ESI-Ion Trap mass spectrometer (Bruker Daltonics, Bremen, Germany). Metabolite separation was accomplished with a Nucleodur Sphinx RP column (250 × 4.6 mm, 5 µm particle size; Macherey–Nagel, Düren, Germany). The mobile phase consisted of 0.2 % formic acid (A) and acetonitrile (B) utilizing a flow of 1 ml $min^{-1}$ with the following gradient: 0 min, 10 % B, 15 min: 55 % B, 15.1 min: 100 % B, 16 min: 100 % B, followed by column reconditioning (*Huber et al., 2015*). To search for unknown metabolites of TA-G, we visually compared the chromatograms of the anterior midgut of latex- and control-fed larvae and subsequently performed tandem mass spectrometry ($MS^2$) experiments using AutoMS/MS runs on the Esquire 6,000 ESI-Ion Trap MS to obtain structure information. Using QuantAnalysis (Bruker Daltonics), TA-G, TA, and the putative TA-GSH conjugates were quantified based on their most abundant ion trace: TA-G: 685 [M+[M-162]], negative mode, retention time (RT) = 12.2 min; TA: 263 [M + H], positive mode, RT = 16.8 min; TA-GSH: 570 [M + H], positive mode, RT = 10.1 min; TA-Cys-glycine: 441 [M + H], positive mode, RT = 9.4 min; TA-Cys-glutamate: 513 [M + H], positive mode, RT = 10.4 min, TA-Cys: 384 [M + H], positive mode, RT = 9.8 min.

## NMR analysis of TA conjugates from *M. melolontha* midgut extract

In order to identify the structures of the putative TA conjugates, we allowed 15 *M. melolontha* larvae to feed for 1 month on *T. officinale* plants. Larvae were then recovered and dipped for 2 s in liquid nitrogen before dissecting them on ice. The entire midgut was homogenized in 1 ml methanol by shaking the samples for 3 min with three metal beads in a paint shaker. The samples were centrifuged at room temperature for 10 min at 17,000 $\times g$, passed through a 0.45 µm cellulose filter, and subsequently purified by high-pressure liquid chromatography (HPLC). NMR analyses were conducted using a 500 MHz Bruker Avance HD spectrometer equipped with a 5 mm TCI cryoprobe. Capillary tubes (2 mm) were used for structure elucidation in MeOH-$d_4$. The analysis revealed the presence of TA-Cys by comparison with a synthesized standard (see below, *Figure 1—figure supplement 2* and *Figure 1— figure supplement 6*). Other TA conjugates identified by high-pressure liquid chromatography-mass spectrometry (HPLC-MS) were below the detection threshold of NMR.

## Synthesis of TA-G metabolite standards for identification and quantification

In order to characterize and quantify the TA-G metabolites, we isolated and synthesized TA-G, TA-G-GSH, TA-G-Cys, TA, TA-GSH, and TA-Cys. TA-G was purified from *T. officinale* latex methanol extracts as described in *Huber et al., 2016b*. TA was obtained by incubating 50 mg purified TA-G with 25 mg β-glucosidase from almonds (Sigma Aldrich) in 2.5 ml $H_2O$ at 25 °C for 2 days. The sample was centrifuged at room temperature for 5 min at 17,000 $\times g$ and supernatant was discarded. The TA-containing pellet was dissolved in 100 µl dimethylsulfoxide (DMSO) and diluted in 1.9 ml 0.01 M TAPS ([tris(hydroxymethyl)methylamino]propanesulfonic acid) buffer (pH = 8.0). Subsequently, solid-phase extraction was performed with a 500 mg HR-X Chromabond cartridge (Macherey-Nagel). The cartridge was washed and conditioned with two volumes of methanol and $H_2O$, respectively. Separation was accomplished using one volume each of $H_2O$, 30 % methanol, and 60 % methanol, and two volumes of 100 % methanol. TA was eluted in the first 100 % methanol fraction, in which no impurities were detected on an Esquire 6,000 ESI-Ion Trap-MS. Samples were evaporated under $N_2$ flow at room temperature to almost complete dryness, and 1 ml $H_2O$ was added before freeze-drying. To obtain TA-GSH and TA-Cys conjugates, the most abundant TA conjugates in the liquid chromatography-mass spectrometry (LC-MS) chromatograms, we dissolved 5 mg isolated TA in 5 µl DMSO in two separate Eppendorf tubes and added 1.6 ml 0.01 M TAPS buffer (pH 8.0) and a 75-fold molar excess of either GSH or Cys to the tubes. Similarly, to obtain TA-G-GSH and TA-G-Cys conjugates, we dissolved 5 mg TA-G in 1 ml 0.01 M TAPS (pH = 8.0) in two separate Eppendorf tubes and added a 75 molar excess of either GSH or Cys. TA-GSH, TA-G-GSH, and TA-G-Cys samples were incubated for 2 days and TA-Cys for 7 days in the dark at 25 °C, after which most of the TA and TA-G had spontaneously conjugated. All samples were stored at –20 °C until purification by semi-preparative HPLC.

Semi-preparative HPLC was accomplished using an HPLC coupled with ultraviolet (HPLC-UV) system coupled to a fraction collector (Advantec SF-2120) using a Nucleodur Sphinx RP column (250 × 4.6 mm, 5 µm particle size; Macherey-Nagel). The mobile phase consisted of 0.01 % formic acid (A) and acetonitrile (B). Flow rate was set to 1 ml min$^{-1}$ with the following gradient: 0 min: 15 % B, 5 min: 30 % B, 9 min: 54 % B, 9.01 min: 100 % B, followed by column reconditioning. Compounds were monitored with a UV detector at 245 nm. As the synthesis resulted in the formation of several isomers that differed in retention times, the conjugates with the same retention times as found in *M. melolontha* larvae were collected. The elution times of the compounds were TA-G-GSH: 6.9 min; TA-G-Cys: 6.4 min; TA-GSH: 8.6 min; TA-Cys: 8.3 min. The fractions were concentrated under nitrogen flow at 30 °C and subsequently lyophilized. The final yields of the conjugates were TA-G-GSH: 2.1 mg; TA-G-Cys: 0.38 mg; TA-GSH: 1.47 mg; TA-Cys: 0.23 mg. Purified fractions were analyzed by NMR spectroscopy for structure verification. Structures with chemical shifts are depicted in Figure 1 - figure supplements 3-6. Standard curves of the conjugates were prepared using 100 µg of the respective compounds in 100 % methanol on an Agilent 1200 HPLC system (Agilent Technologies,) coupled to an API 3200 tandem mass spectrometer (Applied Biosystems, Darmstadt, Germany) equipped with a turbospray ion source operating in negative ionization mode. Injection volume was 5 µl. Metabolite separation was accomplished on a ZORBAX Eclipse XDB-C18 column (50 × 4.6 mm, 1.8 µm; Agilent Technologies). The mobile phase consisted of 0.05 % formic acid (A) and acetonitrile (B) using a flow rate of 1.1 ml min$^{-1}$ with the following gradient: 0 min: 5 % B, 0.5 min: 5 % B, 4 min: 55 % B, 4.1 min:

90 % B, 5 min: 90 % B, followed by column reconditioning. The column temperature was kept at 20 °C. The ion spray voltage was maintained at –4.5 keV. The turbo gas temperature was set at 600 °C. Nebulizing gas was set at 50 psi, curtain gas at 20 psi, heating gas at 60 psi, and collision gas at 5 psi. Multiple reaction monitoring (MRM) in negative mode monitored analyte parent ion → product ion: m/z 423 → 261 (collision energy (CE) –14 V; declustering potential (DP) –40 V) for TA-G; m/z 730 → 143, (CE –66 V; DP –80 V) for TA-G-GSH; m/z 544 → 382 (CE –26 V; DP –80 V) for TA-G-Cys; m/z 261 → 217 (CE –14 V; DP –30 V) for TA; m/z 568 → 143 (CE –44 V; DP –50 V) for TA-GSH; m/z 382 → 120 (CE –30 V; DP –45 V) for TA-Cys; m/z 568 → 143 (CE –44 V; DP –50 V) for loganic acid. Both Q1 and Q3 quadrupoles were maintained at unit resolution. Analyst 1.5 software (Applied Biosystems) was used for data acquisition and processing. Weight-based response factors of TA-G, TA, and their conjugates were calculated relative to loganic acid (Extasynthese, Genay, France). The weight-based response factors were as follows: TA-G: 2.8; TA-G-GSH: 2.5, TA-G-Cys: 1.9; TA: 0.3; TA-GSH: 1.9; TA-Cys: 1.1.

## Quantification of *M. melolontha* TA-G metabolism

In order to quantify the deglucosylation of TA-G and conjugation to GSH, we performed a Waldbauer assay in which we analyzed the TA-G metabolites in *M. melolontha* larvae after consumption of a fixed amount of TA-G. Eight larvae were starved for 7 days before offering them 100 mg of an artificial diet (*Huber et al., 2016b*) supplemented with 100 µg purified TA-G, obtained as described above. Larvae were allowed to feed in the dark for 24 hr in a 180 ml plastic beaker covered with a moist tissue paper, after which the larvae had completely consumed the food. Frass was collected in 500 µl methanol containing 1 µg*ml$^{-1}$ loganic acid as an internal standard. Subsequently, larvae were dipped for 2 s in liquid nitrogen and the anterior midgut, posterior midgut, hindgut content and tissue, and hemolymph and fat tissue removed by dissection. For the gut samples, gut content was collected separately from the gut tissue. All samples were homogenized in 500 µl methanol containing 1 µg*ml$^{-1}$ loganic acid by vigorously shaking the tubes for 2 min with two to three metal beads in a paint shaker. All samples were centrifuged at room temperature for 10 min at 17,000 ×*g*. Supernatants were analyzed by LC-MS on the API 3200 triple quadrupole mass spectrometer as described above using a 5 µl injection volume. Metabolites were quantified based on loganic acid as an internal standard using the Analyst 1.5 software.

## Distribution of TA-Cys upon prolonged exposure of *M. melolontha* to TA-G

To assess the distribution of the major TA-G metabolism product, TA-Cys, in *M. melolontha* exposed for a prolonged time to TA-G, we dissected larvae that were feeding for 1 month on *T. officinale* plants into anterior and posterior midgut, hindgut, fat tissue, skin, and hemolymph as described above. 10 µl hemolymph was collected inside 100 µl methanol. All other tissue samples were homogenized with 10 µl methanol per mg material by vigorously shaking the tubes for 2 min with two to three metal beads in a paint shaker. All samples were centrifuged at room temperature for 15 min at 17,000 ×*g*. Supernatants were analyzed on the HPLC 1100 series equipment coupled to an Esquire 6,000 ESI-Ion Trap mass spectrometer, and the abundance of TA-Cys quantified as described above.

## pH-dependent hydrolysis of TA-G in *T. officinale* latex

In order to test whether TA-G is hydrolyzed by plant enzymes, we analyzed the hydrolysis of TA-G in latex that was extracted in buffers that covered the pH range present in the plant vacuole (pH 5), plant cytosol (pH 7), and *M. melolontha* gut (pH 8) (*Egert et al., 2005*). We cut the main roots of *T. officinale* plants 0.5 cm below the stem-root junction and collected the exuding latex of an entire plant in 1 ml 0.05 M MES (2-(*N*-morpholino)ethanesulfonic acid) buffer (pH 5.2), 0.05 M TRIS-HCl buffer (pH 7.0), or 0.05 M TRIS-HCl (pH 8.0), with three replicates for each buffer. Samples were kept at room temperature for 5 min before stopping the reaction by boiling the samples for 10 min at 98 °C, during which TA-G was found to be stable. Samples were centrifuged at room temperature for 10 min at 17,000 ×*g*, and the supernatant was analyzed by an HPLC 1100 series instrument (Agilent Technologies), coupled to a photodiode array detector (G1315A DAD; Agilent Technologies). Metabolite separation was accomplished as described in *Huber et al., 2015*. Peak areas for TA-G and its aglycone TA were integrated at 245 nm. As the absorption spectra of TA-G and TA do not differ, we expressed the

deglucosylation activity as the ratio of the peak area of TA/(TA + TA G). pH-dependent difference in the deglucosylation activity was analyzed using the Kruskal-Wallis rank sum test.

To investigate the precise pH optimum of the plant hydrolases, and to test for spontaneous hydrolysis of TA-G at acidic pH, we extracted *T. officinale* latex in buffers with a pH range of 3–6. Main root latex was collected as described above, extracted in 2 ml $H_2O$ containing 20 % glycerol, and 200 µl extract was immediately suspended in equal volumes of a series of 0.1 M citrate buffers adjusted to pH 3.0, 3.6, 4.2, 4.8, 5.4, and 6.0. Half of the latex-buffer solution was immediately incubated for 10 min at 95 °C to block enzymatic reaction. The remaining samples were kept at room temperature for 15 min to allow enzymatic reaction and subsequently heated for 10 min at 95 °C. Samples were centrifuged at room temperature at 17,000 ×*g* and the supernatant was analyzed on HPLC-UV as described above. The peak area of TA-G and TA was integrated at 245 nm, and the deglucosylation activity was expressed as TA/(TA + TA -G).

### In vitro deglucosylation of TA-G by *M. melolontha* gut enzymes

In order to test for the presence of TA-G-deglucosylating enzymes in *M. melolontha*, we analyzed the formation of TA in crude extracts of the anterior midgut, posterior midgut, and hindgut. Six *M. melolontha* larvae were starved for 1 week, after which they were cooled for 10 min at –20 °C before dissection. Larvae were dissected into the anterior and posterior midgut and hindgut, with the gut content separated from the gut tissue. Gut samples were weighed and homogenized in 0.01 M TAPS buffer (pH 8.0) containing 10 % glycerol with 10 µl per mg tissue using a plastic pestle. For the deglucosylation assay, 30 µl gut samples that had either been kept on ice or boiled for 10 min at 95 °C were incubated with 30 µl latex extract (prepared as described below) for 20 min at 25 °C, after which the reaction was stopped by heating the samples for 10 min at 95 °C. Samples were centrifuged at 17,000 ×*g* at room temperature for 10 min, after which the supernatant was diluted 1:1 in 0.01 M TAPS buffer (pH 8.0) and stored at –20 °C until chemical analysis. Latex extract was obtained by extracting the entire main root latex of six *T. officinale* plants in 6 ml 0.01 M TAPS buffer (pH = 8.0), after which the samples were immediately heated for 10 min at 95 °C. The latex samples were centrifuged for 20 min at 17,000 ×*g* and filtered through a 0.45 µm cellulose filter. HPLC-UV analysis and quantification of TA-G and TA were carried out as described above. Deglucosylation activity was expressed as the ratio of TA/(TA + TA -G). Differences between the deglucosylation activity of the gut extract and heat treatment were analyzed with a two-way ANOVA.

### Deglucosylation of TA-G by *M. melolontha* in vivo in the absence and presence of plant hydrolases

To test whether *M. melolontha* enzymes are sufficient to deglucosylate TA-G, we fed larvae with a TA-G-supplemented diet that contained *T. officinale* latex extracts that had been left intact or heat deactivated. Eight larvae were starved for 2 weeks before offering them approximately 0.35 cm³ boiled carrot slices coated with 50 µl of intact or heat-deactivated latex extract. Latex extracts were obtained by cutting the main roots of *T. officinale* plants 0.5 cm below the tiller and collecting the latex of an entire plant in 100 µl of either ice-cooled (for intact extracts) or 95 °C (for heat-deactivated extracts) $H_2O$. *M. melolontha* larvae were allowed to feed in the dark inside 180 ml beakers covered with soil for 4 hr. Subsequently, regurgitant was collected in 1 ml methanol by gently prodding the larvae. Left-over food was frozen in liquid nitrogen, ground to a fine powder, and 50 mg ground tissue was extracted with 500 µl methanol by vortexing the samples for 30 s. All samples were centrifuged at room temperature for 10 min at 17,000 ×*g* and the supernatant analyzed by LC-MS on an Esquire 6,000 ESI-Ion Trap-MS (Bruker Daltonics) as described above. TA-G, TA, TA-GSH, and TA-Cys were integrated as described above using QuantAnalysis. Statistical differences in the metabolite abundance between the sample type (food, regurgitant) and the presence of active plant enzymes were analyzed with two-way ANOVAs for each metabolite separately.

### Inhibition of TA-G deglucosylation by *M. melolontha* in vitro

To test whether glucosidases or carboxylesterases mediate the deglucosylation of TA-G, we measured this activity in *M. melolontha* gut extracts in the presence of either carboxylesterase or glucosidase inhibitors. Bis(p-nitrophenyl)phosphate was used as a carboxylesterase inhibitor, whereas castanospermine was deployed as a glucosidase inhibitor that reduces the activity of both α- and β-glucosidases.

Six larvae were starved for 12 days before dissection. The anterior midgut content was extracted in 0.01 M TAPS buffer (pH 8.0) containing 10 % glycerol using 10 µl per mg gut material. To obtain TA-G as a substrate for the deglucosylation assay, the entire main root latex of each of the 15 *T. officinale* plants was collected in 150 µl 0.1 M TAPS (pH 8.0) and samples were immediately heated for 10 min at 95 °C. The samples were centrifuged at room temperature for 10 min at 17,000 ×$g$, and the supernatants were pooled and diluted 1:10 in $H_2O$. The enzymatic assay was performed by incubating 10 µl of the diluted latex TAPS extract with 20 µl gut extract and 30 µl 0, 0.002, or 0.2 mM carboxylesterase or glucosidase inhibitor for 1 hr at room temperature. As a negative control, half volumes of the 0 mM inhibitor samples were immediately incubated at 95 °C to stop the enzymatic reaction. Samples were centrifuged at room temperature for 10 min at 17,000 ×$g$ and the supernatant was analyzed on an HPLC-UV as described above. TA-G and TA were quantified by integrating the peak area at 245 nm. Deglucosylation activity was expressed as the ratio of TA/(TA + TA -G).

To investigate whether α- or β-glucosidases mediate the hydrolysis of TA-G, we measured deglucosylation activity in *M. melolontha* midgut extracts in the presence of acarbose, a specific α-glucosidase inhibitor, or castanospermine, which inhibits both α- and β-glucosidases. Three L3 *M. melolontha* larvae were starved for 5 days, dipped for 2 s in liquid nitrogen, dissected, and the anterior midgut content and tissue extracted in 10 µl 0.15 M NaCl per mg material. Samples were homogenized with a plastic pestle and centrifuged at 4 °C for 10 min at 17,000 ×$g$. Then, 20 µl of the supernatant was incubated with 20 µl boiled latex TAPS extract (obtained as described above) and 0.002, 0.2, or 20 mM acarbose or castanospermine (added in 40 µl) for 1 hr at room temperature. The reaction was stopped by heating for 10 min at 95 °C. Samples were centrifuged at room temperature for 10 min at 17,000 ×$g$ and the supernatant was analyzed on an HPLC-UV as described above. The peak areas of TA-G and TA were integrated at 245 nm. Deglucosylation activity was expressed as the ratio of TA/(TA + TA -G).

## Transcriptome sequencing and analysis

In order to identify the putative *M. melolontha* β-glucosidases, we sequenced 18 anterior and posterior midgut transcriptomes (three treatments, two gut tissues, three replicates of each) from larvae feeding on control, TA-G-enriched, or latex-containing diets using Illumina HiSeq 2,500. 15 *M. melolontha* larvae were starved for 10 days. For 3 consecutive days, larvae were offered 0.35 cm$^3$ boiled carrot slices that were coated with either (i) 50 µl water ('control'), (ii) 50 µl latex water extract that contained heat-deactivated latex of the main root of one *T. officinale* plant ('TA-G enriched'), or (iii) the entire main root latex from one *T. officinale* plant ('latex enriched'). The latex water extract was obtained by collecting the main root latex of 15 *T. officinale* plants in a total of 1.5 ml 95 °C hot water. After 15 min of incubation at 95 °C, the sample was centrifuged at room temperature for 10 min at 17,000 ×$g$ and the supernatant was stored at –20 °C. Food was replaced every day. All larvae consumed at least 95 % of the offered food during the entire period of the experiment. On the third day, the larvae were dissected 4 hr after being fed. Larvae were dipped in liquid nitrogen for 2 s and, subsequently, anterior and posterior midguts were removed by dissection. The gut tissue was cleaned from the gut content, immediately frozen in liquid nitrogen, and stored at –80 °C until RNA extraction. For RNA extraction, gut tissue was ground to a fine powder using plastic pestles. RNA was extracted from 10 to 20 mg ground tissue using innuPREP RNA Mini Kit (Analytik Jena, Jena, Germany) following the manufacturer's protocol. On-column digestion was performed with the innuPREP DNAse I Digest Kit (Analytik Jena). TrueSeq compatible libraries were prepared and PolyA enrichment performed before sequencing the transcriptomes on an Illumina HiSeq 2,500 with 17 Mio reads per library of 100 base pairs, paired-end. Reads were quality trimmed using Sickle with a Phred quality score of >20 and a minimum read length of 80. De novo transcriptome assembly was performed with the pooled reads of all libraries using Trinity (version Trinityrnaseq_r20131110), running at default settings. Raw reads were archived in the NCBI Sequence Read Archive (SRA; BioProject PRJNA728510). Transcript abundance was estimated by mapping the reads of each library to the reference transcriptome using RSEM (*Li and Dewey, 2011*) with Bowtie (version 0.12.9) (*Langmead et al., 2009*) running at default settings. Differential expression analysis was performed with Wald test in DeSeq2 in which low-expressed genes were excluded. Gene ontology (GO) terms were retrieved using Trinotate, and GO enrichment analysis of the up-regulated genes (Benjamini-Hochberg adjusted $p$-value < 0.05) in the anterior midgut of the control and TA-G-enriched samples, as well as the control and latex-enriched samples,

was performed using the hypergeometric test implemented in BiNGO using the Benjamini-Hochberg adjusted p-value of <0.01.

## Identification, phylogenetic, and expression analysis of *M. melolontha* b-glucosidases

In order to identify putative *M. melolontha* β-glucosidases, we performed tBLASTn analysis using the known β-glucosidases from *T. molitor* (AF312017.1) and *C. populi* (KP068701.1) as input sequences (*Ferreira et al., 2001*; *Rahfeld et al., 2015*). We retained transcripts with a BitScore larger than 200, an average FPKM (fragments per kilobase of transcript per million mapped reads) value (all samples) larger than 2, and an at least twofold higher average FPKM value in the anterior than posterior midguts of the control samples to match the in vitro deglucosylation activity. Through this analysis, 19 sequences were selected of which 11 appeared to be full-length genes and 8 were gene fragments.

In order to verify the gene sequences, RNA was isolated from *M. melolontha* anterior midgut samples (three biological replicates) using the RNeasy Plant Mini Kit (Qiagen), and first-strand cDNA was prepared from 1.2 µg of total RNA using SuperScript III reverse transcriptase and oligo d(T$_{12-18}$) primers (Invitrogen, Carlsbad, CA, USA). RACE PCR ('SMARTer RACE cDNA Amplification Kit' Clontech, Mountain View, CA, USA) was used to obtain full-length genes (see *Supplementary file 2* for primer information). In the end, 12 full-length open reading frames of putative β-glucosidases could be amplified from *M. melolontha* cDNA (see *Supplementary file 1* for *M. melolontha* β-glucosidase nucleotide sequences and *Supplementary file 2* for primer information), a reduction from the 19 originally selected sequences due to a lack of amplification of some gene fragments, merging of others, and assembly errors in the transcriptome. Signal peptide prediction of the resulting 12 candidate genes was performed with the online software TargetP (http://www.cbs.dtu.dk/services/TargetP/) (*Emanuelsson et al., 2000*). We aligned the amino acid sequences of the 12 candidate sequences, as well as of the known glucosidases of *T. molitor* (AF312017.1), *C. populi* (KP068701.1), *Brevicoryne brassicae* (AF203780.1), and *Phyllotreta striolata* (KF377833.1) (*Beran et al., 2014*; *Jones et al., 2002*) using the MUSCLE algorithm (gap open, –2.9; gap extend, 0; hydrophobicity multiplier, 1.2; clustering method, upgmb) implemented in MEGA 5.05 (*Tamura et al., 2011*), and visualized the alignment in BioEdit version 7.0.9.0 (*Hall, 1999*). The alignment was used to compute a phylogeny with a maximum likelihood method (WAG model; gamma-distributed rates among sites (five categories); Nearest-Neighbor-Interchange heuristic method; sites with less than 80 % coverage were eliminated) as implemented in MEGA 5.05. A bootstrap resampling strategy with 1000 replicates was applied to calculate tree topology.

In order to estimate the expression levels of the putative β-glucosidases, we replaced the previously identified β-glucosidase sequences in the transcriptome with the confirmed full-length genes and estimated transcript abundance by mapping the trimmed short reads of each library to the corrected reference transcriptome as implemented in the Trinity pipeline using RSEM and Bowtie. For differential expression analysis, all contigs that had an average count value of >1 per library were retained. To test whether TA-G or latex affected the expression of the β-glucosidases, differential expression analysis was accomplished by pairwise comparisons of the control and TA-G-enriched anterior midgut samples, and the control and latex-enriched anterior midgut samples, using an exact test in edgeR (*Robinson et al., 2010*). The significance level of 0.05 was adjusted for multiple testing using the Benjamini-Hochberg false discovery rate method. To test whether the expression level of β-glucosidases differed between anterior and posterior midgut samples, a pairwise comparison between the control samples of the anterior and posterior midgut was performed as described above. Averaged FPKM values of each treatment and gut section were displayed with a heat map.

## Cloning and heterologous expression of *M. melolontha* b-glucosidases

In order to characterize the isolated *M. melolontha* β-glucosidase genes, they were heterologously expressed in a line of *T. ni*-derived cells (High Five Cells; Life Technologies, Carlsbad, CA, USA) as described in *Rahfeld et al., 2015*. Briefly, genes were cloned into the pIB/V5-His TOPO vector (Life Technologies). After sequence verification, these vector constructs were individually used with the FuGeneHD-Kit to transfect insect High Five Cells according to the manufacturer's instructions (Promega, Madison, WI, USA). After 1 day of incubation at 27 °C, the cultures were supplied with 60 mg*ml$^{-1}$ blasticidin (Life Technologies) to initiate the selection of stable cell lines. Afterwards, the

insect cells were selected over three passages. The cultivation of the stable cell lines for protein expression was carried out in 75 cm$^3$ cell culture flasks, containing 10 ml Express Five culture medium (Life Technologies), 20 mg*ml$^{-1}$ blasticidin, and one x Protease Inhibitor HP Mix (SERVA Electrophoresis, Heidelberg, Germany). After 3 days of growth, the supernatant was collected by centrifugation (4000 ×g, 10 min, 4 °C), concentrated using 10.000 Vivaspin 4 (Sartorius), and desalted (NAP-5; GE Healthcare, Munich, Germany) into assay buffer (100 mM NaPi, pH 8).

## Enzymatic assays of recombinant proteins

In order to test the TA-G-hydrolyzing activity and substrate specificity of the *M. melolontha* glucosidases, the heterologously expressed proteins were assayed with the plant defensive glycosides TA-G, a mixture of maize benzoxazinoids (BXDs), salicin, and 4-methylsulfinylbutyl glucosinolate (4-MSOB), as well as the disaccharide cellobiose, which were obtained as described below. The standard fluorogenic substrate, 4-methylumbelliferyl-β-D-glucopyranoside (Glc-MU), served as a positive control. Non-transfected insect cells (WT) and cells transfected with green fluorescent protein (GFP) served as negative controls. For the enzymatic assays, 97 µl concentrated and desalted supernatant of the heterologous expression culture was incubated with 3 µl 10 mM substrate for 24 hr at 25 °C, after which the reaction was stopped with an equal volume of methanol. Due to a very rapid deglucosylation of TA-G, incubation time was shortened to 10 s for this compound. After assays, all samples were centrifuged at 11,000 ×g for 10 min at room temperature and the supernatant was analyzed with a different method for each substrate as described below.

TA-G was purified as described in *Huber et al., 2016b*. Deglucosylation activity was measured based on the concentration of the aglycone TA on an HPLC-UV and quantified at 245 nm as described above.

BXDs were partially purified from maize seedlings (cultivar Delprim hybrid). Seeds were surface-sterilized and germinated in complete darkness. After 20 days, leaves from approximately 60 seedlings were ground under liquid nitrogen to a fine powder and extracted with 0.1 % formic acid in 50 % methanol with 0.25 ml per 100 mg tissue. Methanol was evaporated under nitrogen flow at 40 °C. BXDs were enriched using 500 mg HR-X Chromabond solid-phase extraction cartridges (Macherey-Nagel) with elution steps (5 ml) using water, 30 % (aq.) methanol, and 100 % methanol. 2 ml water was added to the 100 % methanol fraction, which contained the BXDs. Subsequently, methanol was completely evaporated from this fraction under nitrogen flow at 40 °C, and after freeze-drying, the freeze-dried material (~5 mg) was dissolved in 1 ml H$_2$O. This enriched BXD solution contained a mixture of different BXD glucosides, with DIMBOA (2,4-dihydroxy-7-methoxy-1,4-benzoxazin-3-one)-glucoside as the major compound. To test for the deglycosylation of the BXDs, the formation of the aglycone MBOA (6-methoxy-benzoxazolin-2-one; a spontaneous degradation product of the DIMBOA aglycone) was monitored on an Agilent 1200 HPLC system coupled to an API 3200 tandem mass spectrometer (Applied Biosystems) equipped with a turbospray ion source operating in negative ionization mode. Injection volume was 5 µl using a flow rate of 1 ml*min$^{-1}$. Metabolite separation was accomplished with a ZORBAX Eclipse XDB-C18 column (50 × 4.6 mm, 1.8 µm; Agilent Technologies) using the following gradient of 0.05 % formic acid (A) and methanol (B): 0 min: 20 % B, 9 min: 25 % B, 10 min: 50 % B, 12 min: 100 % B, followed by column reconditioning. The column temperature was kept at 20 °C. MRM was used to monitor analyte parent ion → product ion: m/z 164 → 149 (CE –20 V; DP –24 V) for MBOA. Analyst 1.5 software (Applied Biosystems) was used for data acquisition and processing.

Salicin (Alfa Aeser) was purchased and its deglucosylation was quantified based on the formation of the deglucosylation product salicyl alcohol, which was analyzed on an HPLC-UV using the same procedure as described for TA-G. The peak of salicyl alcohol (elution time = 9.3 min) was integrated at 275 nm.

4-MSOB was isolated from 50 g of broccoli seeds (Brokkoli Calabraise; ISP GmbH, Quedlinburg, Germany), which were homogenized in 0.3 l of 80 % aqueous methanol and centrifuged at 2500 ×g for 10 min, and the supernatant separated on a DEAE-Sephadex A25 column (1 g). After the supernatant was loaded, the column was washed three times with 5 ml formic acid + isopropanol + water (3 : 2 : 5 by volume) and four times with 5 ml water. Intact glucosinolates were eluted from the DEAE Sephadex with 25 ml of 0.5 M K$_2$SO$_4$ (containing 3 % isopropanol) dropped into 25 ml of ethanol (*Thies, 1988*). The collected solution was centrifuged to spin down the K$_2$SO$_4$ and the supernatant was dried

under vacuum. The residue was resuspended in 3 ml of water, and 4-MSOB was isolated by HPLC as described in *Schramm et al., 2012*. Purification was performed on an Agilent 1,100 series HPLC system using a Supelcosil LC-18-DB Semi-Prep column (250 × 10 mm, 5 µm; Supelco, Bellefonte, PA, USA) with a gradient of 0.1 % (v/v) aqueous trifluoroacetic acid (solvent A) and acetonitrile (solvent B). Separation was accomplished at a flow rate of 4 ml min$^{-1}$ at 25 °C as follows: 0–3% B (6 min), 3–100% B (0.1 min), a 2.9 -min hold at 100 % B, 100–0% B (0.1 min), and a 3.9 -min hold at 0 % B, and the fraction containing 4-MSOB was collected with a fraction collector. The fraction was dried under vacuum and resuspended in 10 ml methanol to which 40 ml ethanol was added to precipitate the glucosinolate as the potassium salt. The flask was evaporated under vacuum to remove the solvents, and the residue was recovered as a powder. The identity and purity of the isolated 4-MSOB were checked by LC-MS (Bruker Esquire 6000; Bruker Daltonics, Bremen) and 1 H NMR (500 MHz model; Bruker BioSpin GmbH, Karlsruhe, Germany). The deglucosylation of 4-MSOB was quantified based on the formation of 4-MSOB isothiocyanate with an API 3200 LC-MS as described above operating in positive ionization mode. Injection volume was 5 µl using a flow rate of 1.1 ml*min$^{-1}$. Metabolite separation was accomplished with ZORBAX Eclipse XDB-C18 column (50 × 4.6 mm, 1.8 µm; Agilent Technologies) using the following gradient of 0.05 % formic acid (A) and acetonitrile (B): 0 min: 3 % B, 0.5 min: 15 % B, 2.5 min: 85 % B, 2.6 min: 100 % B, 3.5 min: 100 % B, followed by column reconditioning. The column temperature was kept at 20 °C. MRM was used to monitor analyte parent ion → product ion: m/z 178 → 114 (CE –13 V; DP –26 V). Analyst 1.5 software (Applied Biosystems) was used for data acquisition and processing.

Cellobiose (Fluka) was purchased and its deglucosylation was quantified based on the decrease of substrate on an API 3200 LC-MS as described above operating in negative ionization mode. Injection volume was 5 µl, using a flow rate of 1 ml*min$^{-1}$. Metabolite separation was accomplished with an apHera NH$_2$ column (15 cm x 4.6 mm x 3 µm) using the following gradient of H$_2$O (A) and acetonitrile (B): 0 min: 20 % A, 0.5 min: 20 % A, 13 min: 45 % A, 14 min: 20 % A, followed by column reconditioning. The column temperature was kept at 20 °C. MRM was used to monitor analyte parent ion → product ion: m/z 341 → 161 (CE –10 V; DP –25 V). Analyst 1.5 software was used for data acquisition and processing.

Glc-MU (Sigma Aldrich), a fluorogenic substrate, served as a rapid positive control for the presence of β-glucosidases. Hydrolysis of Glc-MU was scored visually by the presence of fluorescence in samples excited with UV light at 360 nm using a gel imaging system (Syngene).

Activity of the heterologously expressed β-glucosidases was categorized into presence and absence based on the formation of the respective aglycones of TA-G, BXDs, salicin, and 4-MSOB, and the decrease of the substrate for cellobiose. For the secondary metabolites, activity was accepted if the aglycone concentration was threefold higher than the mean aglycone concentration of the controls (GFP, WT; except only WT for TA-G). For cellobiose, activity was scored as positive if the cellobiose concentration after the assay was lower than 30 % of the cellobiose concentration of the controls (GFP, WT). The enzymatic assays were performed three times (except TA-G only twice) with freshly harvested recombinant proteins within 2 weeks, which gave similar results ( *Figure 3—figure supplement 2*). The averaged categorization results are displayed in *Figure 3C*.

### *M. melolontha* gut enzymatic assays with plant defensive glycosides

In order to test whether *M. melolontha* gut proteins deglucosylate BXDs, 4-MSOB, and salicin, we tested glucohydrolase activity of crude extracts of the anterior midgut in vitro. 10 *M. melolontha* larvae were starved for 24 h, after which the larvae were dipped for 2 s in liquid nitrogen, and, subsequently, anterior midgut tissue and gut content were removed by dissection. The samples were extracted with 10 µl ice-cold 0.1 M TAPS (pH 8.0) per mg material as described above. All samples were centrifuged at 17,000 ×*g* for 5 min at 4 °C and the supernatant stored at –20 °C until the enzymatic assay. Deglucosylation activity was measured by incubating 20 µl gut extract that had been either kept on ice or boiled for 10 min at 95 °C with a 6 mM mixture of BXDs, salicin, or 4-MSOB (substrates were obtained as described above added in a 20 µl volume) in 0.01 M TAPS (pH 8.0) for 1 hr at room temperature, after which the reaction was stopped by the addition of an equal volume of methanol. All samples were centrifuged at 3220 ×*g* for 5 min at room temperature and the supernatant stored at –20 °C until analysis. For BXDs, salicin, and 4-MSOB, the formation of the aglycone was quantified using HPLC-MS and HPLC-UV as described above. Deglycosylation activity

was standardized by dividing the peak area of the aglycone of each sample by the maximal peak area of all samples ('relative aglycone formation'). Differences in the relative aglycone formation between boiled and non-boiled gut samples, as well as between anterior midgut content and tissue samples, were analyzed with two-way ANOVAs.

## Development of RNAi methodology for *M. melolontha* larvae

In order to establish RNAi in *M. melolontha*, we injected different doses of dsRNA targeting *tubulin* and *GFP* (negative control) into the larvae. As a template for dsRNA synthesis, we chose an approximately 500 bp fragment of each gene (see **Supplementary file 3** for fragment nucleotide sequence). The fragments were amplified using the Q5 High-Fidelity DNA Polymerase (New England Biolabs, Ispwich, MA, USA) according to the manufacturer's procedure and the specific primer combinations Mm-tubulin-fwd and Mm-tubulin-rev for *tubulin*, as well as GFP-RNAi_fwd and GFP-RNAi_rev for *GFP* (**Supplementary file 2**). Isolated and purified *M. melolontha* cDNA served as a template for *tubulin*. Plasmid pGJ 2648, which encodes for the emerald variant for *GFP* and was kindly supplied by Dr. Christian Schulze-Gronover, served as a template for *GFP*. Amplified fragments were separated by agarose gel electrophoresis and purified using GeneJET Gel Extraction Kit (Thermo Fisher Scientific, Waltham, MA, USA) according to the manufacturer's procedure. An A-tail was added using DreamTaq DNA Polymerase (Thermo Fisher Scientific), and the A-tailed fragments were then cloned into T7 promoter sequence containing pCR2.1-TOPO plasmids (Life Technologies) according to the manufacturer's instructions. Plasmids with the insert in both orientations with regard to the T7 promoter were identified by sequencing.

dsRNA was synthesized using the MEGAscript RNAi Kit (Thermo Fisher Scientific) according to the manufacturer's procedure. The above-described tubulin and GFP plasmid templates were linearized downstream of the insert using the restriction enzyme BamHI (New England Biolabs). Sense and antisense single-stranded (ss) RNAs were synthesized in separate reactions. The complementary RNA molecules were then annealed and purified using MEGAscript RNAi Kit according to the manufacturer's instructions (Thermo Fisher Scientific).

In order to investigate the required dsRNA concentration and duration of the silencing, we injected 2.5 and 0.25 µg dsRNA of *tubulin* or *GFP* per g of larva into *M. melolontha*. The larvae were anesthetized under $CO_2$. Subsequently, larvae were punctured with a sterile syringe (Ø 0.30 × 12 mm) between the second and the third segment, and approximately 50 µl *tubulin* or *GFP* dsRNA (100 ng*µl$^{-1}$ for 2.5 µg per g larva and 10 ng*µl$^{-1}$ for 0.25 µg per g larva) was injected into the hemolymph of the second segment of nine *M. melolontha* larvae per concentration. Every second day, the larvae were weighed. 5 days after injection, the larvae received fresh carrots to feed on. 2, 5, and 10 days after injection, three larvae per concentration were frozen in liquid nitrogen. The entire larvae were ground to a fine powder using mortar and pestle under liquid nitrogen and stored at –80 °C until RNA extraction. Total RNA was isolated using the GeneJET Plant RNA Purification Kit following the manufacturer's instructions. On-column RNA digestion was performed with RNase-free DNase (Qiagen, Netherlands). cDNA synthesis was performed using SuperScript II Reverse Transcriptase (Thermo Fisher) and oligo (dT$_{21}$) (Microsynth, Switzerland) according to the manufacturer's instructions. Consequently, the qPCR was performed with the KAPA SYBR FAST qPCR Kit Optimized for LightCycler 480 (Kapa Biosystems, Wilmington, MA, USA) in a Nunc 96-well plate (Thermo Fisher Scientific) on a LightCycler 96 (Roche Diagnostics, Switzerland) with one technical replicate per sample. *Tubulin* gene expression was quantified relative to actin using the qPCR primers qPCR_Mm_Tubulin_fwd and qPCR_Mm_Tubulin_rev for *tubulin*, as well as qPCR_Mm_actin_fwd and qPCR_Mm_actin_rev for *actin* (**Supplementary file 2**). Differences in the relative expression of *tubulin* to *actin* and between *tubulin*- and *GFP* dsRNA-treated larvae were analyzed with the Student's t-test.

## Synthesis of dsRNA for RNAi

In order to test whether Mm_bGlc17 accounts for the TA-G deglucosylation in vivo, we silenced *Mm_bGlc16*, *Mm_bGlc17*, and *Mm_bGlc18* in *M. melolontha* using RNAi and analyzed TA-G deglucosylation activity in vitro using anterior midgut extracts. *M. melolontha* in which a *dsGFP* fragment was injected served as a control. *GFP* dsRNA was synthesized as described above. To obtain dsRNA for the glucosidase genes, we chose approximately 500 bp fragments of *Mm_bGlc16*, *Mm_bGlc17*, and *Mm_bGlc18* cDNA as templates for dsRNA synthesis that showed maximal sequence divergence

with other *M. melolontha* β-glucosidases as well as among each other (see **Supplementary file 3** for fragment nucleotide sequence). The fragments were amplified using the Q5 High-Fidelity DNA Polymerase (New England Biolabs) according to the manufacturer's procedure and specific primer combinations of which one primer was fused to the T7 promoter sequence. The plasmids obtained from the heterologous expression were used as PCR templates (see above). For each β-glucosidase, we performed two PCRs to yield two dsRNA templates that are identical except for a single T7 promoter sequence at opposite ends. For *Mm_bGlc16* fragment amplification, the primer combinations Mm_bGlc_16_fwd_T7 and Mm_bGlc_16_rev, as well as Mm_bGlc_16_fwd and Mm_bGlc_16_rev_T7, were used. For the amplification of *Mm_bGlc17* and *Mm_bGlc18* fragments, the respective primers were deployed. Amplified fragments were separated by agarose gel electrophoresis and purified using GeneJET Gel Extraction Kit (Thermo Fisher Scientific) according to the manufacturer's procedure. An A-tail was added using DreamTaq DNA Polymerase (Thermo Fisher Scientific) and the A-tailed fragments were then cloned into pIB/V5-His-TOPO plasmids. dsRNA was synthesized and linearized as described above using the restriction enzymes XhoI, for the glucosidase genes, and BamHI, for GFP (New England Biolabs). The dsRNA was synthesized using the MEGAscript RNAi Kit (Thermo Fisher Scientific) according to the manufacturer's procedure. The above-described *M. melolontha* β-glucosidase and *GFP* plasmid templates were linearized downstream of the insert using restriction enzymes XhoI and BamHI (New England Biolabs), respectively, and annealed and purified as described above.

## TA-G deglucosylation activity in RNAi-silenced *M. melolontha* larvae

To silence *M. melolontha* glucosidases in vivo, we injected dsRNA of the respective glucosidases or *GFP* as a control into *M. melolontha* larvae as described above using 50 µl of a 10 ng*µl$^{-1}$ *Mm_bGlc16*, *Mm_bGlc17*, *Mm_bGlc18*, or *GFP* dsRNA. In addition, we performed a co-silencing of *Mm_bGlc16* and *Mm_bGlc17 (Mm_bGlc16&17)*, for which 25 µl of 10 ng*µl$^{-1}$ *Mm_bGlc16* and *Mm_bGlc17* was injected. Larvae were kept at room temperature for 7 days, after which the larvae were dissected as described above. The anterior midgut content was extracted with 10 µl 0.01 M TAPS (pH 8.0) per mg material and centrifuged at 17,000 ×*g* for 10 min at 4 °C. For the enzymatic reaction, 10 µl supernatant that was either kept at 4 °C or had been boiled for 1 hr at 98 °C was incubated with 40 µl 0.01 M TAPS (pH 8.0) and 50 µl 2 mM latex water extract. After 3 hr, the reaction was stopped by adding equal volumes of methanol. The samples were centrifuged at 17,000 ×*g* for 10 min at room temperature and the supernatant analyzed on a Waters ACQUITY UPLC series equipment coupled to an ACQUITY photodiode array and an ACQUITY QDa mass detector. Metabolite separation was accomplished using an ACQUITY UPLC column with 1.7 µm BEH C18 particles (2.1 × 100 mm). The mobile phase consisted of 0.05 % formic acid (A) and acetonitrile (B) utilizing a flow rate of 0.4 ml*min$^{-1}$ with the following gradient: 0 min: 5 % B, 1.5 min: 20 % B, 2.5 min: 40 % B, 3 min: 95 % B, 5 min: 95 % B, followed by column reconditioning. The peak areas of TA and TA-G were integrated at 245 nm using Waters MassLynx[49]. Deglucosylation activity was expressed as the ratio of TA/(TA + TA -G). In addition, to account for the spontaneous deglucosylation of TA-G, the deglucosylation activity was normalized by subtracting the average TA/(TA + TA -G) of the boiled samples from each non-boiled sample ('normalized deglucosylation activity'). Difference in the normalized and non-normalized deglucosylation activities between the RNAi-silenced larvae was analyzed with one-way ANOVAs, and significant differences between the groups were determined using Tukey's Honest Significance test.

## *Mm_bGlc17* silencing efficiency and specificity

To test for the silencing efficiency and specificity of the *Mm_bGlc17* dsRNA injection, we injected *M. melolontha* with 0.25 µg of *Mm_bGlc17* or *GFP* dsRNA per g larva as described above. Non-injected larvae were set as controls. After injections, larvae were kept at room temperature for 2 days, after which the larvae were dissected and the individual midguts were isolated. Then, total RNA of the midgut was extracted using RNeasy Lipid Tissue Mini Kit (QIAGEN), coupled with on-column DNA digestion following the manufacturer's instructions. One microgram of each total RNA sample was reverse transcribed with SuperScript III Reverse Transcriptase (Invitrogen). The quantitative reverse transcription PCR (RT-qPCR) assay (N = 7–8) was performed on the LightCycler 96 Instrument (Roche) using the KAPA SYBR FAST qPCR Master Mix (Kapa Biosystems). The *actin* gene was used as an internal standard to normalize cDNA concentrations. The relative gene expressions of *Mm_bGlc16*, *Mm_bGlc17*, and *Mm_bGlc18* to *actin* were calculated with $2^{-\Delta\Delta Ct}$ method. Primers (qPCR_Mm

_bGlc_16_fwd, qPCR_Mm _bGlc_16_rev, qPCR_Mm _bGlc_17_fwd, qPCR_Mm _bGlc_17_rev, qPCR_ Mm _bGlc_18_fwd, qPCR_Mm _bGlc_18_rev, qPCR_Mm _actin-fwd, and qPCR_Mm _actin-rev) are listed in *Supplementary file 2*.

## Effects of *Mm_bGlc17* silencing on *M. melolontha* performance

In order to test whether Mm_bGlc17 activity affects the performance of *M. melolontha* larvae in the presence and absence of TA-G, we assessed the growth of *Mm_bGlc17*- and control (*GFP*)-silenced larvae on TA-G-deficient and control *T. officinale* plants. *T. officinale* seeds were germinated on seedling substrate. After 15 days, plants were transplanted into 1 l rectangular pots (18 × 12 × 5 cm, length × width × height) filled with a homogenized mixture of 2/3 seedling substrate (Klasmann-Deilmann, Switzerland) and 1/3 landerde (Ricoter, Switzerland). Each pot consisted of four plants in two parallel rows of two plants, which were arranged along the short edges of the pots. Rows were spaced 9 cm apart and had a distance of 4.5 cm from the short edges, and plants within each row were grown 4 cm apart from each other. After 50 days of growth, half of the pots (N = 15 per genotype) were randomly selected to examine the performance of *Mm_bGlc17*-silenced larva, and the second half of the pots (N = 15 per genotype) were used for *GFP*-control larva. dsRNA was synthesized as described above. Larvae were treated with 0.25 µg of *Mm_bGlc17* or *GFP* dsRNA per g larva as previously described. One pre-weighed larva was added into a hole (4 cm depth, 1 cm diameter) in the center of the pots and covered with moist soil. After 3 weeks of infestation, larvae were recovered from the pots, reweighed, and the midgut was extracted for subsequent RNA extraction following the above-mentioned protocol. To reduce the possible effects of environmental heterogeneity within the greenhouse, the position and direction of the pots were randomly re-arranged weekly. Total RNA of the midgut was extracted using RNeasy Lipid Tissue Mini Kit (QIAGEN), coupled with on-column DNA digestion following the manufacturer's instructions. One microgram of each total RNA sample was reverse transcribed with SuperScript III Reverse Transcriptase (Invitrogen). The RT-qPCR assay was performed on the LightCycler 96 Instrument (Roche) using the KAPA SYBR FAST qPCR Master Mix (Kapa Biosystems). The *actin* gene was used as an internal standard to normalize cDNA concentrations. The relative gene expressions to *actin* were calculated with $2^{-\Delta\Delta Ct}$ method.

Differences in *M. melolontha* weight gain between larval and plant RNAi treatments were analyzed with a two-way ANOVA. Differences in larval weight gain between *Mm_bGlc17*-silenced and *GFP*-control larvae were analyzed with Student's t-tests for larvae grown on wild-type and TA-G-deficient plants separately. Differences in larval weight gain on TA-G-containing and TA-G-lacking *T. officinale* plants were analyzed with Student's t-tests for the *Mm_bGlc17*-silenced and *GFP*-control larvae separately. A two-way ANOVA was applied to analyze differences in relative *Mm_bGlc17* expression between larval and plant RNAi treatments. Relative *Mm_bGlc17* expression was thereto log-transformed to improve model assumptions. Differences in relative *Mm_bGlc17* expression between larvae growing on TA-G-containing and TA-G-lacking plants were analyzed with Kruskal-Wallis rank sum tests based on untransformed data for *Mm_bGlc17*-silenced and *GFP*-control larvae separately.

To repeat the above-described experiment, *T.officinale* seeds of TA-G-deficient and control plants were cultivated in the greenhouse as previously described, with some slight modifications. Seedlings were germinated on seedling substrate and transplanted into individual pots (11 × 11 x 11 cm) after 21 days of growth (N = 40 per line). After 70 days of growth, larvae were treated with 0.25 µg of *Mm_bGlc17* or *GFP* dsRNA per g larva as described above. 4 days later, for each *T. officinale* line, half of the plants were infested with one pre-weighed *Mm_bGlc17*-silenced larva and the other half was infested with one pre-weighed *GFP*-control larva. After 3 weeks of infestation, larvae were carefully recaptured from the pots, weighed, and added into the pots again. 5 weeks later, larvae were recaptured again and weighed.

Differences in *M. melolontha* weight gain between larval and plant RNAi treatments were analyzed with two-way ANOVAs for three time periods (3 weeks, 3–8 weeks, and 8 weeks after the start of the experiment) separately. Differences in larval weight gain between *Mm_bGlc17*-silenced and *GFP*-control larvae in these three time periods were analyzed with Student's t-tests for wild-type and TA-G-deficient monocultures separately.

## Effects of *Mm_bGlc17* silencing on deterrence of TA-G

In order to test whether *M. melolontha* glucosidase activity affects the deterrence of TA-G, we assessed the choice of *Mm_bGlc17*- and control (*GFP*)-silenced larvae between TA-G-deficient and control *T. officinale* plants. *M. melolontha* larvae were injected with 0.025 µg*g$^{-1}$ *Mm_bGlc17* or *GFP* dsRNA as described above. 1 week after dsRNA injection, the larvae were starved for 3 days and placed individually into the center of 250 ml plastic beakers filled with vermiculite. 5-week-old TA-G-deficient and control *T. officinale* seedlings were embedded into the vermiculite-filled beaker at opposite edges, with 37 replicated beakers for each of the *Mm_bGlc17* and *GFP* treatments. The feeding site was scored visually 3 hr after the start of the experiment by inspecting the beakers from outside. Differences in the choice between TA-G-deficient and control *T. officinale* plants were analyzed with binomial tests for the *Mm_bGlc17*- and *GFP*-silenced larvae separately.

## Acknowledgements

We would like to thank Daniel Giddings-Vassão, Verena Jeschke, Nataly Wilsch, Ilham Sbaiti, Katharina Lüthy, Zohra Aziz, and Ines Cambra for supporting the experimental procedures, as well as Tobias Köllner for help in the sequence analyses and Dr. Christian Schulze-Gronover for providing plasmids. The project was funded by the Max-Planck Society, the German Research Foundation to MH (project number 422213951), the Swiss National Science Foundation to MH (P400PB_186770), the Seventh Framework Programme for Research and Technological Development of the European Union to ME (FP7 MC-CIG 629134), and the University of Münster.

## Additional information

### Funding

| Funder | Grant reference number | Author |
|---|---|---|
| Deutsche Forschungsgemeinschaft | 422213951 | Meret Huber |
| Schweizerischer Nationalfonds zur Förderung der Wissenschaftlichen Forschung | P400PB_186770 | Meret Huber |
| FP7 Research and Technological Development of the European Union | FP7 MC-CIG 629134 | Matthias Erb |
| Schweizerischer Nationalfonds | 310030_189071 | Christelle AM Robert |
| Schweizerischer Nationalfonds | 153517 | Matthias Erb |

The funders had no role in study design, data collection and interpretation, or the decision to submit the work for publication.

### Author contributions

Meret Huber, Conceptualization, Formal analysis, Funding acquisition, Investigation, Methodology, Project administration, Resources, Supervision, Validation, Visualization, Writing – original draft, Writing – review and editing; Thomas Roder, Alexander Riedel, Formal analysis, Investigation, Methodology, Writing – review and editing; Sandra Irmisch, Julia Fricke, Formal analysis, Investigation, Methodology, Supervision, Writing – original draft, Writing – review and editing; Saskia Gablenz, Formal analysis, Investigation; Peter Rahfeld, Methodology, Supervision, Writing – review and editing; Michael Reichelt, Formal analysis, Methodology, Writing – review and editing; Christian Paetz, Formal analysis, Investigation, Methodology, Visualization, Writing – review and editing; Nicole Liechti, Formal analysis, Writing – review and editing; Lingfei Hu, Ye Meng, Wei Huang, Christelle AM Robert,

Investigation, Writing – review and editing; Zoe Bont, Formal analysis, Investigation, Writing – review and editing; Jonathan Gershenzon, Funding acquisition, Project administration, Resources, Supervision, Writing – review and editing; Matthias Erb, Conceptualization, Formal analysis, Funding acquisition, Methodology, Project administration, Resources, Supervision, Validation, Writing – original draft, Writing – review and editing

### Author ORCIDs
Meret Huber (ID) http://orcid.org/0000-0002-8708-394X
Thomas Roder (ID) http://orcid.org/0000-0002-6642-6288
Michael Reichelt (ID) http://orcid.org/0000-0002-6691-6500
Christelle AM Robert (ID) http://orcid.org/0000-0003-3415-2371
Jonathan Gershenzon (ID) http://orcid.org/0000-0002-1812-1551
Matthias Erb (ID) http://orcid.org/0000-0002-4446-9834

### Decision letter and Author response
Decision letter https://doi.org/10.7554/eLife.68642.sa1
Author response https://doi.org/10.7554/eLife.68642.sa2

## Additional files

### Supplementary files
• Supplementary file 1. Nucleotide sequences of the heterologously expressed *M. melolontha* β-glucosidases.

• Supplementary file 2. Primer information.

• Supplementary file 3. Fragment nucleotide sequence of *tubulin*, *GFP*, *Mm_bGlc16*, *Mm_bGlc17*, and *Mm_bGlc18* in *pCR2.1-TOPO* and *pIB/V5-His-TOPO* vectors.

• Transparent reporting form

### Data availability
Sequencing data have been deposited in small read archive SRA under BioProject PRJNA728510. All data generated during this study are included in the manuscript and supporting files. Source data files have been provided for all figures and figures supplements if appropriate.

The following dataset was generated:

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
