## [Decision Letter]

**Acceptance summary:**

This paper is of interest to the broader audience interested in the coevolutionary arms race between plants and their herbivores. In a thoroughly investigated case study the detoxification strategy of cockchafer larvae towards the major defensive compound of one of their preferred host plants, dandelion, are revealed and effects on the behavior of the larvae described.

**Decision letter after peer review:**

Thank you for submitting your article "A β-glucosidase of an insect herbivore determines both toxicity and deterrence of a dandelion defense metabolite" for consideration by *eLife*. Your article has been reviewed by 3 peer reviewers, including Youngsung Joo as the Reviewing Editor and Reviewer #1, and the evaluation has been overseen by Meredith Schuman as the Senior Editor. The following individual involved in review of your submission has agreed to reveal their identity: Susanne Dobler (Reviewer #2).

Essential revisions:

1) Both reviewer 1 and 3 argue that the current discussion for the deterrent behavior causing by silencing β-glucosidase need to be more specified.

2) Detoxification of TA-G by cockchafer larvae has been nicely analyzed in Figure 1, but there are more interesting results which the author did not properly interpret, e.g. the frass only contained a trace amount of TA-associated metabolites. Both reviewers 2 and 3 argue that the metabolic fate of TA-G needs to be discussed more.

*Reviewer #1 (Recommendations for the authors):*

However, I still consider that the following concerns should be considered and addressed:

1. I think one of the key take-home messages of this manuscript is the behavioral consequence of herbivore digestive enzymes. It is a very interesting and striking result. I know it would be very difficult to give a solid mechanism for how it works, but it is necessary to discuss how Mm_bGluc17-associated deglycosylation can alter herbivore behavior. In addition, to explain the feeding preference data, the authors suggested two hypotheses. I fully agree the first one ("good feeding site hypothesis"), but I could not understand well for the second hypothesis. Could you explain which ecological context really Mm_bGluc17-dependent cleavage of TA-G decrease herbivore fitness and then why they still have Mm_bGluc17-dependent cleavage of TA-G?

2. Inducibility of Mm_bGluc17 is inconsistent between Figure 3 and Figure 4. In Figure 3, transcript abundances of Mm_bGluc17 did not differ among the treatment groups (water, TA-G, and Latex). However, the transcript abundances of Mm_bGluc17 decreased in the larvae fed on TA-G deficient plants compared to the larvae fed on WT plants. Could you explain why they are different?

Reviewer #3 (Recommendations for the authors):

1. Regarding the metabolic fate of dietary TA-G, a major metabolite was detected as TA-Cys (25%), which is distributed in the body and mainly accumulated in the anterior midgut (Lines 160 – 163 and Figure 1D), whereas only a trace amount of TA-G was identified in frass, compared to the total amount consumed within 24 hr (Lines 163-164 and Lines 176 – 177, Figure 1E). It leads me to call into a couple of questions: (1) Is TA-Cys the ultimate final metabolite of TA-G? Or can it be further processed to other metabolites? (2) It seems that TA-Cys is not likely excreted out of body. If it continues, the larvae could be doused with a thick concentration of TA-Cys. Do you think it is sequestered for any further utilization for its own benefit? Otherwise, it should be ultimately excreted, which doesn't seem so within 24 hr. (3) The distribution map is a snapshot at 24 hr post-feeding. Can it be a different distribution pattern in a long-term observation? Please give your answers either to each question or as a whole. I suggest authors add a paragraph in Discussion to deal with this issue. Current discussion is mostly focused on the β-glucosidase found in the herbivore gut, but I think the metabolic fate data would pave a ground way for further research.

2. As soon as TA-G is hydrolyzed, it seems to be simultaneously conjugated with GSH, resulting in TA-Cys after consecutive enzymatic processes. Although β-glucosidase is the enzyme acting at the first step, other enzymes, such as glutathione S-transferase (GST) and peptidases, may not be less significant. Non-enzymatic processes could be carried out instead, since it seems to happen very quickly. However, I cannot find any mention about these possible occasions. I suggest authors add a mention probably together with the comment 1 above.

3. Regarding the second evidence of your claim (Lines 345-347), TA-G is deglycosylated by the gut extracts not only in the absence of plant material, but also in the presence of plant material (Figure 2C – TA-G panel). Also, "the presence of plant extract with active TA-G glycohydrolases does not increase TA-G deglycosylation", compared to what? It looks true when compared to the insect gut extracts only, but it is not true when compared to the treatment of "plant enzyme + and diet" (Figure 2C – TA panel). Also the "active TA-G glycohydrolases" is not clear what it means: plant or insect one? From another perspective, I am wondering whether the reaction substrate (TA-G) was sufficiently enough or not in the latex coated on carrot slices for the enzymatic conversions. The Figure 2C – TA-G panel shows that the substrate TA-G was almost completely depleted (presumably by deglycosylation) at the given condition (substrate concentration, reaction time, etc.), which could probably tell that the TA-G was not sufficiently enough to in the beginning. If there was a sufficient amount of substrate, it could have been shown that "the plant extract with active TA-G glycohydrolases" DOES increase TA-G deglycosylation in addition to the insect activity. How would you think? Anyways, the Lines 345-347 should be more clearly rephrased.

4. Regarding the deterrent effect, it seems very interesting that silencing a detoxification enzyme has nullified the deterrent behavior (Figure 4D). Although authors try to propose two reasonable mechanisms, I think it remains still elusive. I am wondering if there is any chemosensory-mediated mechanism to be postulated. In Line 375, the sentence, "the enzyme also prompts M. melolontha larvae to avoid TA-G", sounds a bit excessive, that is required to be tone down and rephrased.

5. Both deglucosylation and deglycosylation are mix used throughout the manuscript. If there is no special emphasis or intention, it would look better to use a single consistent term. Likewise, Glu, Glc and Gluc are also mix used especially for the Mm_bGlc gene/enzyme names. For examples, Mm_bGlu18 (Line 276), Mm_bGluc17 (Lines 278 and 279; Figure 4B-D; Line 402), and probably more in other places.

6. Figure 2B Is there any special reason to draw the bar graph, instead of box-plots (as shown in Figures 2A and 2C)?

7. Figure 3C Why don't you place TA-G at the first column? TA-G is your main focus.

8. Line 366 "plant neighbors" should be replaced by a clear and definitive term, or referred to certain groups of organisms in the cited paper. Also "good feeding sites" (Line 377) is also a vague expression. Rephrase it.

---

## [Author Response]

Essential revisions:1) Both reviewer 1 and 3 argue that the current discussion for the deterrent behavior causing by silencing β-glucosidase need to be more specified.2) Detoxification of TA-G by cockchafer larvae has been nicely analyzed in Figure 1, but there are more interesting results which the author did not properly interpret, e.g. the frass only contained a trace amount of TA-associated metabolites. Both reviewers 2 and 3 argue that the metabolic fate of TA-G needs to be discussed more.

Thank you for summarizing these points. We now address these issues in the revised manuscript, and replied to the reviewers point by point.

Reviewer #1 (Recommendations for the authors):However, I still consider that the following concerns should be considered and addressed:1. I think one of the key take-home messages of this manuscript is the behavioral consequence of herbivore digestive enzymes. It is a very interesting and striking result. I know it would be very difficult to give a solid mechanism for how it works, but it is necessary to discuss how Mm_bGluc17-associated deglycosylation can alter herbivore behavior.

Thank you for this suggestion. The neurophysiological mechanisms that lead to TA-G avoidance are unclear. On the one hand, it is possible that TA or a TA conjugates bind to specific receptors inside the gut, and thereby modulate herbivore behaviour. On the other hand, TA may be present at low concentrations in the larval mouth part, either through regurgitants or the presence of Mm_bGlc17 in the insect saliva, and activate gustatory receptors that alter herbivore behavior. We now discuss these possibilities in more detail.

In addition, to explain the feeding preference data, the authors suggested two hypotheses. I fully agree the first one ("good feeding site hypothesis"), but I could not understand well for the second hypothesis. Could you explain which ecological context really Mm_bGluc17-dependent cleavage of TA-G decrease herbivore fitness and then why they still have Mm_bGluc17-dependent cleavage of TA-G?

While the glucohydrolysis of TA-G seems to improve *M. melolontha* growth under laboratory conditions, the presence of other biotic or abiotic stresses may reverse the beneficial effects of TA-G deglycosylation. For instance, the hydrolysis of TA-G may deplete the level of cysteine inside the larva through conjugation to glutathione. Under stressful conditions, e.g. under nutrient limitations or in the presence of other toxic allelochemicals, the depletion of cysteine may have negative impacts on the larva. If Mm_bGlc17 is however critical for the larvae to acquire nutrients, the gene may still be maintained in the genome, as the net effect might be positive for the larva. Thus, the common dandelion would exploit a promiscuous β-glucosidase whose evolution is constrained by its primary function in nutrient acquisition. We now explain these points in more details in the discussion.

2. Inducibility of Mm_bGluc17 is inconsistent between Figure 3 and Figure 4. In Figure 3, transcript abundances of Mm_bGluc17 did not differ among the treatment groups (water, TA-G, and Latex). However, the transcript abundances of Mm_bGluc17 decreased in the larvae fed on TA-G deficient plants compared to the larvae fed on WT plants. Could you explain why they are different?

There are several possibilities that may account for the observed differences in the inducibility of Mm_bGlc17 by TA-G. First, this difference might be due to different methodologies and sample sizes used (RNAseq, N=3 vs qPCR, N=12-14). qPCR experiments with high samples sizes are more powerful in detecting differences in gene expression among treatment groups than RNAseq experiments with small sample sizes. Second, the differences in the inducibility may be caused by different TA-G concentrations, the presence of a natural plant matrix, or the genotypic background of the *M. melolontha* larvae. We now mention these aspects in the discussion.

Reviewer #3 (Recommendations for the authors):1. Regarding the metabolic fate of dietary TA-G, a major metabolite was detected as TA-Cys (25%), which is distributed in the body and mainly accumulated in the anterior midgut (Lines 160 – 163 and Figure 1D), whereas only a trace amount of TA-G was identified in frass, compared to the total amount consumed within 24 hr (Lines 163-164 and Lines 176 – 177, Figure 1E). It leads me to call into a couple of questions: (1) Is TA-Cys the ultimate final metabolite of TA-G? Or can it be further processed to other metabolites? (2) It seems that TA-Cys is not likely excreted out of body. If it continues, the larvae could be doused with a thick concentration of TA-Cys. Do you think it is sequestered for any further utilization for its own benefit? Otherwise, it should be ultimately excreted, which doesn't seem so within 24 hr. (3) The distribution map is a snapshot at 24 hr post-feeding. Can it be a different distribution pattern in a long-term observation? Please give your answers either to each question or as a whole. I suggest authors add a paragraph in Discussion to deal with this issue. Current discussion is mostly focused on the β-glucosidase found in the herbivore gut, but I think the metabolic fate data would pave a ground way for further research.

To address these questions, we analysed the accumulation of TA-Cys in *M. melolontha* larvae feeding for one month on *T. officinale* plants. This experiment showed that the anterior midgut remains the body part with highest concentration of TA-Cys , followed by the posterior midgut. The hindgut and also the fat tissue contained only small amounts of TA-Cys. Thus, the distribution pattern of *M. melolontha* feeding for prolonged time on TA-G containing diet is similar to the 24h snapshot. Consequently, TA-Cys must be further metabolized to yet unknown products, and these products may either be sequestered or excreted. As we cannot differentiate between these possibilities, we can only speculate whether *M. melolontha* uses TA for its own benefits. We added the data on long-term feeding of *M. melolontha* on *T. officinale* plants into the manuscript and included a paragraph to discuss the metabolic fate of TA-G inside the larva in more detail.

2. As soon as TA-G is hydrolyzed, it seems to be simultaneously conjugated with GSH, resulting in TA-Cys after consecutive enzymatic processes. Although β-glucosidase is the enzyme acting at the first step, other enzymes, such as glutathione S-transferase (GST) and peptidases, may not be less significant. Non-enzymatic processes could be carried out instead, since it seems to happen very quickly. However, I cannot find any mention about these possible occasions. I suggest authors add a mention probably together with the comment 1 above.

We now discuss the possibility of enzymatic and non-enzymatic conjugation of GSH to TA-G in the same paragraph as mentioned above.

3. Regarding the second evidence of your claim (Lines 345-347), TA-G is deglycosylated by the gut extracts not only in the absence of plant material, but also in the presence of plant material (Figure 2C – TA-G panel). Also, "the presence of plant extract with active TA-G glycohydrolases does not increase TA-G deglycosylation", compared to what? It looks true when compared to the insect gut extracts only, but it is not true when compared to the treatment of "plant enzyme + and diet" (Figure 2C – TA panel). Also the "active TA-G glycohydrolases" is not clear what it means: plant or insect one? From another perspective, I am wondering whether the reaction substrate (TA-G) was sufficiently enough or not in the latex coated on carrot slices for the enzymatic conversions. The Figure 2C – TA-G panel shows that the substrate TA-G was almost completely depleted (presumably by deglycosylation) at the given condition (substrate concentration, reaction time, etc.), which could probably tell that the TA-G was not sufficiently enough to in the beginning. If there was a sufficient amount of substrate, it could have been shown that "the plant extract with active TA-G glycohydrolases" DOES increase TA-G deglycosylation in addition to the insect activity. How would you think? Anyways, the Lines 345-347 should be more clearly rephrased.

It is true that the substrate TA-G was depleted after ingestion by the *M. melolontha* larvae, and thus, we cannot exclude that plant hydrolases may contribute to deglycosylation if higher amounts of TA-G are present. However, considering that we added half of the entire root latex of a plant to the diet, which was consumed to a large part, it is very unlikely that in nature the larva is exposed to higher amounts of TA-G than what we tested. Increasing substrate concentration thus would not provide much insight into the ecological relevance of plant-mediated deglycosylation. In addition, our RNAi approach clearly shows that plant hydrolases *in planta* are not sufficient to trigger TA-G avoidance in *M. melolontha*, which is the strongest possible support that insect rather than plant glucohydrolases mediate TA-G cleavage and avoidance behaviour of the larvae in vivo.

To make these aspects clearer, we now improved the description of the experiment and results. We rephrased lines 345-347 to clarify our experiment and conclusions, and describe the results from this experiment in more detail.

4. Regarding the deterrent effect, it seems very interesting that silencing a detoxification enzyme has nullified the deterrent behavior (Figure 4D). Although authors try to propose two reasonable mechanisms, I think it remains still elusive. I am wondering if there is any chemosensory-mediated mechanism to be postulated. In Line 375, the sentence, "the enzyme also prompts M. melolontha larvae to avoid TA-G", sounds a bit excessive, that is required to be tone down and rephrased.

We added a paragraph in the discussion about possible chemosensory mechanisms, see also comment of reviewer 1. We also rephrased the mentioned sentence.

5. Both deglucosylation and deglycosylation are mix used throughout the manuscript. If there is no special emphasis or intention, it would look better to use a single consistent term. Likewise, Glu, Glc and Gluc are also mix used especially for the Mm_bGlc gene/enzyme names. For examples, Mm_bGlu18 (Line 276), Mm_bGluc17 (Lines 278 and 279; Figure 4B-D; Line 402), and probably more in other places.

We now use “deglucosylation” when referring to glucohydolysis, and “deglycosylation” if sugars other than glucose may be cleaved. The term “Glc“ is now used for all gene and enzyme names.

6. Figure 2B Is there any special reason to draw the bar graph, instead of box-plots (as shown in Figures 2A and 2C)?

The focus of Figure 2A and 2B are the difference in the mean deglucosylation activity, which is best visualized using barplots. Consequently, we converted the boxplots of Figure 2A to barplots. In contrast, in Figure 2C, not only the difference but also the variability of the data might be of interest (see comment of reviewer 2); thus, we prefer boxplots as in the original figure.

7. Figure 3C Why don't you place TA-G at the first column? TA-G is your main focus.

We used the positive control as the first column, as a negative activity towards TA-G is uninformative if the positive control is negative. We now highlight in the figure that the first column is the positive control (Glc-MU – pos. ctr).

8. Line 366 "plant neighbors" should be replaced by a clear and definitive term, or referred to certain groups of organisms in the cited paper. Also "good feeding sites" (Line 377) is also a vague expression. Rephrase it.

Done.